# LANGUAGE AND EXPERIENCE: A COMPUTATIONAL MODEL OF SOCIAL LEARNING IN COMPLEX TASKS

**Cédric Colas**
MIT

**Tracey Mills**
MIT

**Ben Prystawski**
Stanford University

**Michael Henry Tessler**
Google DeepMind

**Noah Goodman**
Stanford University

**Jacob Andreas**
MIT

**Joshua Tenenbaum**
MIT

## ABSTRACT

The ability to combine linguistic guidance from others with direct experience is central to human development, enabling safe and rapid learning in new environments. How do people integrate these two sources of knowledge, and how might AI systems? We present a computational framework that models human social learning as joint probabilistic inference over structured, executable world models given sensorimotor and linguistic data. We make this possible by turning a pretrained language model into a probabilistic model of how humans share advice conditioned on their beliefs, allowing our agents both to generate advice for others and to interpret linguistic input as evidence during Bayesian inference. Using behavioral experiments and simulations across 10 video games, we show how linguistic guidance can shape exploration and accelerate learning by reducing risky interactions and speeding up key discoveries in both humans and models. We further explore how knowledge can accumulate across generations through iterated learning experiments and demonstrate successful knowledge transfer between humans and models — revealing how structured, language-compatible representations might facilitate human-machine collaborative learning.

**Code:** github.com/ccolas/language_and_experience
**Demo:** cedriccolas.com/demos/language_and_experience

## 1 INTRODUCTION

Imagine learning to forage mushrooms in autumn woods. Each outing provides direct experience of promising slopes and soil conditions. But an experienced forager's advice can transform your exploration in two crucial ways: "*Never touch the red ones with white spots; they're deadly*" helps you avoid fatal mistakes, while "*Look for chanterelles near oak trees after warm rains*" turns random wandering into focused search. This ability to integrate linguistic guidance with direct experience is fundamental to human intelligence, enabling not only safer and more efficient learning, but also the accumulation of knowledge across generations (Tomasello, 2009; Boyd et al., 2011). Yet, we still lack a general computational account of how humans combine these two modes of knowledge acquisition to inform exploration and decision-making in complex tasks.

Current computational models capture only fragments of this dual learning capability. Reinforcement learning (RL) agents, for example, can master complex tasks but require extensive trial-and-error — millions of interaction steps — before achieving proficiency (Sutton and Barto, 2018; Mnih et al., 2015). Theory-based RL mitigates this limitation by combining planning with Bayesian inference over structured world models, achieving human-like sample efficiency, yet remains incapable of social learning (Tsividis et al., 2021; Griffiths et al., 2010; Lake et al., 2017). Attempts to bridge this gap with language-conditioned RL integrate linguistic input into the learning process but rely on massive amounts of paired experience and language, making real-world application impractical (Zhong et al., 2020; Luketina et al., 2019). While large language models (LLMs) excel at processing linguistic guidance flexibly (Brown et al., 2020), they struggle with interactive planning and embodied learning (Valmeekam et al., 2023; Paglieri et al., 2024). Bayesian models of social cognition provide

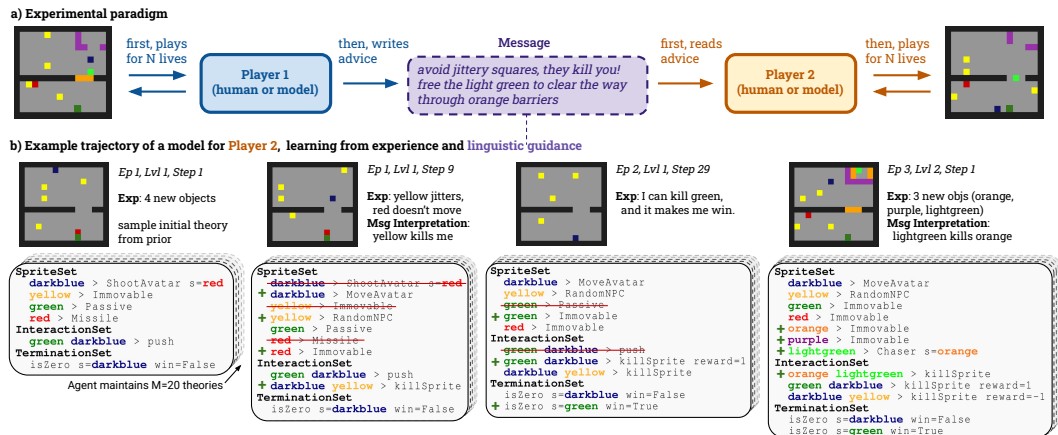

Figure 1: **Overview of the model and experiments** a) *Experimental design*: players (participants or models) are given N=10 lives to learn to play a new video game, either from experience only (Player 1) or from experience and advice written by a previous player (Player 2). b) *Example learning trajectory*: The model maintains beliefs about possible game rules and objectives (programs at bottom), and constantly refines them based on an initial linguistic guidance and new incoming experience.

a promising step towards integrating observed behaviors and language to infer goals (Shafto et al., 2014; Jara-Ettinger et al., 2016), yet they typically operate in simple, non-interactive tasks with predefined hypothesis spaces (Ying et al., 2023; Zhi-Xuan et al., 2024).[1]

Our primary aim is to contribute to cognitive science by offering a computational account of how humans integrate language and experience. We propose a Bayesian framework that treats linguistic guidance and direct experience as complementary evidence sources learners can leverage to infer executable, program-like world models. To make this possible, we introduce three key contributions:

- **LMs as speaker models:** We leverage LMs to approximate the probability that a human with specific world beliefs would produce particular linguistic advice, enabling our model to interpret and generate human-interpretable guidance.

- **Inference from linguistic evidence:** This speaker model is used to evaluate the plausibility of received advice under different candidate world models, allowing linguistic input to shape Bayesian inference alongside experiential learning.

- **LM-accelerated inference:** We use LMs to transform advice into targeted proposal distributions, guiding Bayesian updates towards the most promising regions of the hypothesis space.

These mechanisms allow our computational model to learn efficiently from naturalistic linguistic input, update beliefs in real-time, and share discoveries with others — forming the basis for more structured and communicative learning systems.

We validate our framework through human experiments and computational simulations, showing that linguistic guidance accelerates learning, shapes exploration strategies, and supports knowledge transfer across generations as well as between humans and models. These results reveal how language drives cultural transmission and illustrate how structured, language-compatible representations can facilitate human–AI collaborative learning.

## 2 VIDEO GAMES AS LEARNING ENVIRONMENTS

Video games provide an ideal experimental paradigm for studying social and individual learning (Allen et al., 2024): they offer rich causal environments for systematic exploration, create natural pressure to learn efficiently through costs like lost lives, and enable the study of how complex mechanics can be communicated through language. Drawing inspiration from earlier work modeling human causal learning (Tsividis et al., 2021; Tomov et al., 2023), and cultural knowledge transmission (Tessler et al., 2021), we reuse a suite of 10 games defined in the Video Game Description Language (VGDL) (Schaul, 2013).

---

[1]See more detailed related work in Appendix Section A.

VGDL lets us define games by specifying object types, collision effects, rewards, and win/loss conditions (e.g. yellow objects are deadly, victory requires eliminating green objects; see Figure 2 and DSL description in Appendix Section C). This formal representation enables precise experimental control and provides a structured language for our model to represent and update hypotheses about game dynamics. Players, however, encounter the games as grids of colored squares, having to discover game dynamics and objectives through exploration. Our games were designed to test diverse learning capabilities: some require spatial reasoning to navigate hazards (*pushBoulders*) or use teleportation mechanics (*portals*); others demand quick tactical decisions like shooting threats (*aliens*) or defending resources (*plaqueAttack*). Most challenging are games requiring systematic experimentation to discover novel objects through object combinations (*relational*). Play them **here**.

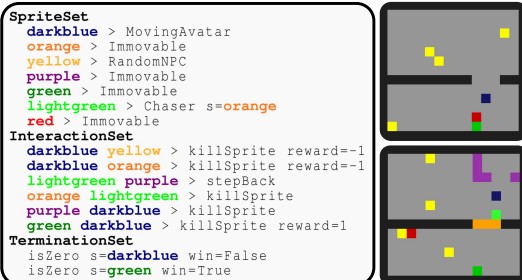

Figure 2: **Game example (beesAndBirds).** Players must discover rules and objectives on their own, sometimes helped with advice from others, to solve 4 game levels (right: levels 1 and 2/4).

## 3 COMPUTATIONAL MODEL OF INTEGRATED LEARNING

We frame game learning as a problem of sequential decision-making under uncertainty (Kaelbling et al., 1998). At each time step $t$, agents select actions $a_t$ to maximize expected cumulative rewards:

$$a_t = \arg\max_a \mathbb{E}_{s_{i+1} \sim P(s_{i+1} \mid s_i, a)} \left[ \sum_{i=t}^{\infty} \gamma^{i-t} r_i \right],$$

where $s_i$ and $r_i$ are the agent's state and reward at time $i$, and $\gamma$ is a discount factor. The main challenge lies in the agent's uncertainty about the transition function $P(s_{i+1} \mid s_i, a)$, which is governed by unknown game dynamics.

Our approach extends the *theory-based RL* framework by proposing to infer causal world models *jointly from experience and linguistic guidance* to support goal-directed planning and strategic exploration (Tsividis et al., 2021). The model alternates between three phases: 1) it constructs and infers a posterior over probabilistic world models given both experience and linguistic guidance, 2) it plans action sequences by identifying and pursuing high-value interactions that balance exploration and exploitation, and (3) it executes these actions in the game. The following sections detail each component of this learning loop.

### 3.1 INFERENCE OF STRUCTURED, CAUSAL WORLD MODELS

Our agent models its environment with a distribution over structured world models, each represented as a probabilistic program specifying game rules and objectives. These beliefs are continually updated as the agent gathers new evidence from gameplay *experience E* and *linguistic guidance L*. We formalize these beliefs update as a Bayesian inference over possible world theories $T$:

$$P(T \mid E, L) \propto P(E \mid T) \times P(L \mid T) \times P(T),$$

where $P(T)$ encodes prior beliefs over plausible world models, while $P(E \mid T)$ and $P(L \mid T)$ measure the consistency of theory $T$ with experiential and linguistic evidence, respectively. As illustrated in Figure 1b, the agent continuously refines its beliefs, incrementally integrating new data from experience and reinterpreting language to narrow down its hypothesis space. In the following subsections, we detail: 1) the search space of possible world models, 2) the likelihood functions that quantify the fit of experience and linguistic guidance, and 3) the inference algorithm that efficiently approximates the posterior distribution over possible worlds.

**A space of possible worlds.** World models are programs specifying the transition function and reward structure, while inference consists of updating a posterior distribution over these executable programs. Each candidate world model, or theory $T$, is represented as a VGDL program that specifies object types for each object color (e.g. missile, shooting avatar), interaction effects between objects

(e.g. collision with yellow kills avatar), reward functions (e.g. +1 when avatar kills green), and win/loss conditions (e.g. kill all green), see example in Figure 2 and complete list of VGDL primitives in Appendix Section C. We define a simplicity-biased prior $P(T)$ over the search space, favoring theories with fewer rules: 1) object types are uniformly distributed, 2) any object pair has a $p = 0.25$ chance of interacting, with interaction types sampled uniformly, and 3) each object's death has a $p = 0.1$ chance of contributing to win or loss conditions. Sampling a theory from this prior involves generating object types, object pair interactions, and win/loss conditions accordingly. These theories are executable: they can be compiled into playable games that lets the agent simulate trajectories internally. Play them yourself on our demo website.

**Likelihood from experience.** To estimate the likelihood $P(E \mid T)$, we first decompose the agent's experience $E$ — a sequence of symbolic state transitions — into a sequence of discrete events $e_i$ (object movements, appearance or disappearance, rewards, and win/loss events). We assume that these local events are conditionally independent given the theory $T$, which lets us factorize the likelihood as $P(E \mid T) = \prod_{i=1}^{n} P(e_i \mid T)$. Because candidate world models are executable, we can estimate $P(e_i \mid T)$ through simulations. Specifically, we replay the agent's actions under $T$ by initializing the game engine to the agent's previous state and executing its chosen action. We then track the occurrence of each observed event $e_i$ across 20 independent simulations, using their frequencies as empirical estimates of $P(e_i \mid T)$.

**Likelihood from language.** Linguistic advice received from other agents serves as evidence for evaluating candidate theories $T$, modeled through *Bayesian Theory-of-Mind* (Baker et al., 2011). We formalize $P(L \mid T)$ as the probability that a speaker, believing $T$ to be true, would produce the observed message $L$. To approximate this, we use a language model (`LLaMA-3.1-70B`) as a probabilistic speaker model. Given a description of $T$, the LM is prompted to generate advice for a future player, see prompt in Appendix J. The likelihood is then estimated as the LM's probability of producing the exact message $L$: $P(L \mid T) \approx P_{\text{LM}}(L \mid \text{prompt}(T))$. This approximation measures how well $T$ explains the speaker's linguistic behavior: e.g. if $L$ contains the advice "avoid yellow at all cost!", then $P(L \mid T)$ will be higher if $T$ contains the rule "yellow kills avatar," than if it does not.

Although $P_{\text{LM}}(L \mid \text{prompt}(T))$ is not an accurate model of human speakers, our inference procedure relies only on relative likelihoods across theories. What matters for the posterior is the pattern of likelihood differences between candidate theories, not the absolute scale. This use of approximate generative models is common in Bayesian cognitive modeling, where the goal is to capture how linguistic evidence shifts beliefs across hypotheses rather than to recover exact speaker probabilities.

**Inference algorithm.** The space of possible theories is vast — exceeding $10^{20}$ configurations for games with just five objects — making exact Bayesian inference intractable. To approximate the posterior distribution $P(T \mid E, L)$, we use a particle filter with Metropolis rejuvenations (Metropolis et al., 1953; Chopin, 2002). We maintain a population of $M = 20$ candidate theories and iteratively refine them by: (1) resampling theories proportional to their posterior probability, and (2) proposing local modifications guided by observed events and linguistic guidance (see details in Appendix Section E). These modify exactly one rule at a time: an object's type, the interaction between a pair of objects, a win condition, a loss condition, or a reward function. They are accepted in proportion to the ratio of posterior probabilities between modified and original theories: $p_{\text{accept}} = min(1, P(T' \mid E, L)/P(T \mid E, L))$ (Metropolis et al., 1953). This process efficiently approximates the posterior distribution $P(T \mid E, L)$ over candidate theories, with each particle $T_i$ assigned a weight $w_i$ proportional to its posterior probability given experience and linguistic guidance. The resulting distribution represents the agent's belief over possible game dynamics and objectives. The agent executes one inference step (1 resampling step + 5 rejuvenation steps) every 20 environment steps, and 20 inference steps every time a new kind of object appears in the scene, or when the agent dies; see full pseudo-code of the inference algorithm in Appendix Section D.

**Language-guided proposals.** We use the same LM to bias the proposal of game rules — e.g. after receiving the message "*yellow kills you,*" the LM may propose rules capturing this lethal interaction:

$$P(r_{\text{new}} \mid r_{\text{old}}, E, L, T) \propto \big((P_0(r_{\text{new}} \mid E, T) + P_{\text{LM}}(r_{\text{new}} \mid \text{prompt}(L, T)))/2$$

where $r_{\text{new}}$ is a candidate rule (e.g. "yellow kills avatar"), $r_{\text{old}}$ is the current rule, $P_0$ is the base proposal distribution, and $P_{\text{LM}}$ is the probability the language model assigns to that rule given the message. This is implemented by prompting the LM with the received message $L$ and instructing it to answer multiple-choice questions about specific VGDL rules: e.g. does the yellow object: 1)

kills the avatar, 2) steps back against the avatar, etc.; see detailed prompt in Appendix Section J. This process biases inference towards theories containing rules most compatible with received advice, resulting in faster convergence.

## 3.2 GOAL-DIRECTED PLANNING AND STRATEGIC EXPLORATION

To maximize its expected long-term utility, the agent must balance *exploration* — gathering information to refine its world model — and *exploitation* — leveraging its current understanding to achieve game objectives. Planning is guided by the maximum a posteriori (MAP) theory $T_{\text{MAP}}$, inferred during Bayesian updates: the agent's best estimate of game rules and objectives. Based on $T_{\text{MAP}}$ simulations, the agent selects high-level goals, and plans action sequences to achieve them.

**Goal sampling.** Based on $T_{\text{MAP}}$, the agent defines a space of high-level goals as object-object interactions that it can cause in the environment: collisions between the agent, something it can push or shoot, and any other object. Each goal is assigned two values: 1) an *exploitation value* reflecting its contribution to game objectives (e.g. higher if it is thought to trigger a reward or a win), and 2) an *exploration value* representing its potential to reduce model uncertainty, measured as the disagreement about what would happen across the $M = 20$ candidate world models. Subgoals are sampled in proportion to their combined value, balancing both learning and game progress.

**Action planning.** To achieve these goals, the agent optimizes 10-step action sequences using $T_{\text{MAP}}$ for simulation. Initial action plans are refined through a simple genetic algorithm to maximize both game rewards and progress toward goals. To prevent catastrophic errors, the agent performs ten 3-step lookaheads to detect possible deaths or major deviations from the expected reward, triggering replanning when necessary. More details about planning can be found in Appendix Section F.

## 3.3 GENERATING LINGUISTIC GUIDANCE FOR OTHERS

The agent generates linguistic advice for future players by sampling from the same speaker model used to evaluate language likelihood during inference — effectively translating its MAP theory $T_{\text{MAP}}$ into natural language $L_{\text{generated}} \sim P_{\text{LM}}(L, |, \text{prompt}(T_{\text{MAP}}))$. This provides optimal speaker modeling when message emitters are computational agents and approximate modeling when they are human. By using the LM both to interpret linguistic guidance and to generate it, the agent captures key aspects of Bayesian Theory of Mind — modeling how humans communicate their beliefs and how they interpret the beliefs of others through language.

## 3.4 BASELINE MODELS

We compare our approach to three baselines: 1) *Oracle*: a model that plans to solve the game using ground-truth game rules, 2) *Deep RL*: a Double Deep Q-Network agent implementing pure trial-and-error learning without structured representations (Mnih et al., 2015; Van Hasselt et al., 2016), and 3) *pure LM*: an LM agent (*LLaMA-3.1-70B*) that leverages state-of-the-art techniques to scaffold long-term decision making: ReAct approach (Yao et al., 2022), use of a scratch pad (Nye et al., 2021) and chain-of-thought reasoning towards beliefs updating and plan formation (Wei et al., 2022); see detailed prompt in Appendix Section J). These baselines test the importance of structured representations and goal-directed planning. Note that we do not compare to language-conditioned RL baselines (Luketina et al., 2019; Colas et al., 2022). These methods rely on thousands of episodes of paired (state, language) supervision to learn how to map linguistic inputs to action policies, and they do not generalize to new, idiosyncratic messages seen only once. In our design, each player receives a single novel message per game — mirroring human one-shot social learning — so language-conditioned RL would receive no opportunity to learn from language and would behave identically to pure deep RL, which we already include as a baseline.

## 4 EXPERIMENTAL PARADIGM

To investigate how humans and computational models integrate experiential and linguistic evidence during learning, we conducted a series of IRB-approved experiments comparing three learning conditions (see Figure 1a):

1. **Experience**: Players learn solely through direct interaction with the game.
2. **Experience + human message**: Players receive additional advice from previous human players.
3. **Experience + model message**: Players receive additional advice from previous model players.

This design allows us to examine both the effectiveness of linguistic guidance and potential asymmetries in human-model knowledge transfer. In each condition, players had 15 lives to solve four levels of each game, advancing only after completing the current level.

**Participants.** We recruited 122 participants through Prolific to play 5 randomly-assigned games. To ensure task engagement while maintaining a representative sample, we excluded participants who failed to complete at least one level in $\geq 3$ games (final N=120). Participants were randomly assigned to one of the 3 conditions (N=40 each). In social conditions, each participant received advice from a randomly-selected previous player (either human or model) who had completed the game in the experience-only condition (1-to-1 mapping).

**Procedure.** All participants first completed a brief tutorial game to familiarize themselves with the interface and basic game mechanics. In social conditions, participants read advice from previous players before starting each new game, and during gameplay. After completing each game (either by winning or depleting lives), participants wrote advice "*to help future players who have not yet played the game.*" This prompt encouraged participants to distill their learned knowledge into linguistic guidance, see full instructions in Appendix Section G.

**Analysis approach.** To evaluate learning efficiency, we tracked both the number of lives required to complete each level and the total proportion of levels completed. We use a normalized area-under-curve (nAUC) metric ($\in [0, 1]$) to integrate both proficiency and learning efficiency. To analyze message effectiveness, we manually coded advice content along four dimensions: the fraction of useful information about game dynamics, about loss conditions, about win conditions, and the presence of incorrect information. This coding scheme lets us examine how specific types of linguistic guidance shape exploration and learning outcomes. We will report differences between conditions as $\Delta$(nAUC).

**Computational simulations.** We conducted 20 simulation runs per condition, matching the human sample size to enable direct comparison. The model received the same information as human participants: in social conditions, it processed the same messages (human- or model-generated) that humans received, while in the experience-only condition, it learned purely through interaction. We also ran iterated learning experiments where a sequence of 10 agents, each given two lives, played the game and passed a message to the next agent. This design tests whether partial knowledge can accumulate incrementally across generations, mirroring human cultural learning (Tessler et al., 2021).

## 5  RESULTS

We first examine how humans and models learn novel games from pure experience, before analyzing how linguistic guidance shapes this process. We then look at human–model transfer, before exploring how our model can accumulate partial knowledge across generations. Our results reveal both striking similarities and systematic differences in learning strategies across humans and models.

### 5.1  LEARNING FROM EXPERIENCE

How efficiently can humans and models learn novel game dynamics through pure experience? Both demonstrated remarkable sample efficiency, with median participants solving 9 of 10 games and our model solving all 10 games within a 10-life budget (Figure 3). However, systematic differences emerged in games requiring specific cognitive capabilities (see per-game plots in Appendix Figure 7). In *relational*, which demands systematic exploration of object combinations, humans showed a bimodal pattern: 25% achieved model-like efficiency by systematically testing interactions, while 40% failed to solve even two out of

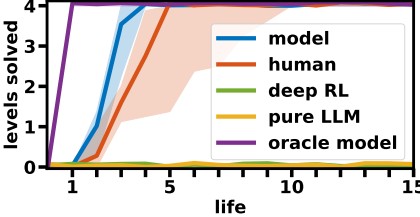

Figure 3: **Learning from experience.** Median across games (N=10, IQ range).

four levels. This split suggests that while humans can perform systematic experimentation, not everyone defaults to it — unlike our model which explicitly reasons about information gain. Conversely, in *avoidGeorge*, which requires rapid planning to protect allies, models consistently outperformed humans (median levels: 4 vs 0), likely due to their capacity for accurate short-term planning.

The importance of structured reasoning becomes clear when comparing against baselines. Pure deep RL (double DQN) failed to solve any level within 10 lives, while pure LM agents never solved more than one level per game, often solving none (7/10 games). The stark difference between these baselines and both human and model performance underscores the value of structured theories of game dynamics in supporting efficient exploration and decision-making. Knowing the model of the world (oracle model) lets agents solve most games within their first life (see Figure 3 and Appendix Figure 7).

## 5.2 LEARNING FROM EXPERIENCE AND HUMAN LANGUAGE

Having established baseline learning capabilities, we next examine how linguistic guidance from previous human players shapes exploration and learning outcomes. Our results demonstrate substantial benefits from social learning while revealing key patterns in effective knowledge transmission.

**Benefits of linguistic guidance.** How does linguistic guidance shape learning? Both humans and our model showed significantly faster learning when provided with human-written advice, reducing median attempts needed by 1.75 for humans (4→2.25) and 1.25 for models (2.5→1.25) (see Figure 4). To quantify these benefits in terms of learning speed, we computed a normalized area under the learning curve (nAUC). A fixed-effect model controlling for game difficulty revealed significant improvements from linguistic guidance for both humans ($\Delta$(nAUC) $= 0.12$, $p = 2.2 \times 10^{-4}$) and models ($\Delta$(nAUC) $= 0.04$, $p = 3.3 \times 10^{-2}$). These benefits were most pronounced in games with opaque mechanics like *relational*, or multiple hazards like *portals* and *jaws*, where guidance could directly communicate critical interactions. Benefits were smallest in games requiring primarily motor skills like *aliens*, *missileCommand* or *plaqueAttack*.

**What makes effective guidance?** Analysis of message content revealed systematic patterns in communication and message effectiveness. Most messages (88%) contained information about game dynamics ("[avoidGeorge] *light blue can transform green into them*"), with many also including information about win (64%, e.g. "[jaws] *The objective is to stay alive*") and loss conditions (74%, e.g. "[avoidGeorge] *you lose if all squares get turned purple*"). A small but notable fraction (11%) contained errors (e.g. "[relational] *Push all the blue into the orange*," when blue should contact yellow). Longer messages and those containing detailed win conditions proved particularly beneficial to both models and humans($p = 6.3 \times 10^{-3}$ and $p = 0.031$ respectively). Importantly, messages that helped humans also helped models, shown by significant correlations in performance gains across nAUC ($r = 0.17$), lives to first level ($r = 0.24$), and lives to second level ($r = 0.16$, all $p < 0.02$) — suggesting shared mechanisms for integrating linguistic and experiential knowledge.

**How language shapes exploration.** Linguistic guidance systematically altered how players explored game mechanics. Messages warning about dangers significantly reduced costly mistakes: in average, humans and models receiving such warnings experienced between 37% and 67% fewer deaths in *avoidGeorge*, *relational*, *jaws*, and *aliens*. Messages about key mechanics accelerated their discovery: in *relational*, players informed about tool creation discovered essential combinations 43% to 83%

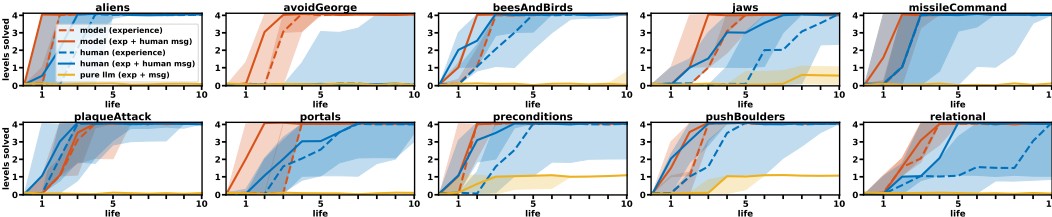

Figure 4: **Learning from experience and human advice.** In most games, both humans (blue) and our model (orange) learn significantly faster with human advice (plain lines) than without (dashed lines), see median in Figure 5a. The *pure LLM* baseline does not learn significantly better with advice, solving the first level in only 3/10 games (N=20, median $\pm$ interquartile range).

faster, while in *plaqueAttack* they learned to revive allies 43% to 62% faster when told about this possibility. However, incorrect advice could also mislead: in *avoidGeorge*, the human player and the model who were wrongly warned about the danger of green blocks (in fact harmless) completely avoided them for the first three episodes, while others players interacted with them in average 5 times in the same period — demonstrating how unhelpful linguistic advice can also shape direct experience.

### 5.3 FROM COGNITIVE MODELS TO LEARNING PARTNERS?

Our model demonstrated the ability to efficiently learn from human-generated advice. But can it also help humans in return? Can it generate guidance that is useful to other models and, more importantly, to humans? Leveraging its LM-based advice generation capability (Section 3.3), our model produced detailed, pedagogical advice that rivaled human teaching. For example, in *preconditions*:

> "*Control the darkblue square with arrow keys. Your goal is to kill all gold objects by touching them, earning points along the way. Watch out for green objects — touching them will kill you unless you have white resources to protect yourself. Collect white resources to safeguard against green, and use them to kill green objects if needed, but be aware that each kill will cost a resource.*"

Model-generated guidance significantly improved learning for both humans ($\Delta$(nAUC)= 0.15, $p < 10^{-5}$) and models ($\Delta$(nAUC)= 0.088, $p < 10^{-8}$) compared to experience alone. Interestingly, models learned better from model- than human-generated advice ($\Delta$(nAUC)= 0.052, $p < 10^{-10}$), while these gave humans only a modest advantage ($\Delta$(nAUC)= 0.035, $p = 0.26$, n.s.).

These asymmetries reveal important differences in human and model communication. When we had our model generate advice based on human players' trajectories, models still learned better from this model-generated guidance than from human-written messages ($\Delta$(nAUC)= 0.21, $p < 10^{-8}$). This suggests the asymmetry stems not from differences in knowledge, but from communication style. Human messages often included metacognitive strategies ("*take time to look for safe patterns*"), analogies ("*orange is like terminator*"), or emotional content ("*[I] was left very confused*") — aspects that human learners readily use but our model finds harder to interpret. These differences highlight the potential and challenges for language-mediated human-machine collaborative learning. We show more examples of human- and model-generated messages in Appendix Section I and on our **website**.

Ablating language-guided proposals in our model leads to a significant performance drop compared to the full version ($\Delta$(nAUC) $= -0.058$, $p = 7.2 \times 10^{-4}$). This ablated version still leverages language likelihood, which lets it outperform learning from experience alone ($\Delta$(nAUC) $= -0.030$, $p = 5.7 \times 10^{-3}$), see Method Section 3.1 and Appendix Figure 8.

**Variability across games:** Our results reveal interesting patterns in the success and failures of linguistic guidance (Figure 5b). Models learning from other models show consistent benefits across all games because models generate more comprehensive messages and process advice more reliably than humans on average. In contrast, human learners show minimal improvements in games requiring rapid reactions and precise motor control (*portals*, *plaqueAttack*, *aliens*, *missileCommand*), where linguistic advice cannot substitute for motor practice. Models struggle to learn from humans in

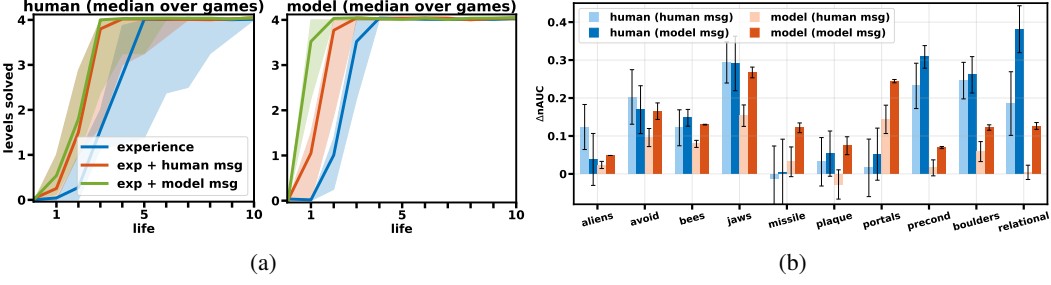

(a)  (b)

Figure 5: **Bidirectional human–model learning.** a) Median performance across games for humans and models learning from experience alone (blue), or experience combined with human-generated (orange) or model-generated (green) linguistic guidance (N=10, shaded: interquartile range). b) Performance improvements over individual learning baselines for each game (error bar: sem).

*relational*, where humans often provide imprecise descriptions of complex rule interactions, and *plaqueAttack*, where humans frequently omit important mechanics they discovered. In contrast, all learners benefit from both human and model advice in games like *avoidGeorge*, where the critical strategy is non-obvious, and *beesAndBirds*, *jaws*, and *pushBoulders*, where key dangers are memorable and straightforward to describe. These patterns suggest that social learning strategies like selecting pedagogical teachers or aggregating multiple sources of advice could mitigate some failure cases.

### 5.4 GENERATIONAL LEARNING

While the previous experiments allowed social learners to benefit from fully explored game mechanics, real-world learning often relies on partial and imperfect knowledge transmission. To investigate whether our model could replicate this gradual accumulation of knowledge, we designed an iterated learning experiment inspired by Tessler et al. (2021). In this setting, each agent interacts with the game environment for only two lives before generating advice to the next agent. This cycle continues across 10 generations for each of the 10 games, with performance tracked generation by generation.

Our results show that performance reliably increases across generations in all games where models do not already achieve mastery from generation 1 (*preconditions* and *aliens*) (Figure 6). Some games reached near-complete mastery by Generation 2, while others showed more gradual improvements. A fixed-effect model controlling for game variability confirmed this trend, showing significant improvements over the first generation for all others ($\Delta(\text{nAUC})_{i>1} \in [0.44, 0.57]$, all $p < 10^{-10}$).

However, in *plaqueAttack* and *relational*, we observe occasional regressions where later generations underperform earlier ones, highlighting the brittleness of single-teacher transmission. These regressions stem from structural properties of the games. Relational requires coordinating many interdependent transformation rules, and even when players know the correct rules, a single mis-push can irreversibly block progress within the two-life limit, making performance volatile across generations. PlaqueAttack involves fast-paced action with two different viable survival strategies (eliminating attackers or reconquering damaged bases). Messages can describe only one of these strategies, which can inadvertently steer later generations away from exploring the alternative, resulting in intermittent drops in performance. These task-specific constraints explain why cumulative improvement is less stable in these games than in others with more linguistically compressible mechanics. This phenomenon could be mitigated through the integration of multiple teachers or teacher selection (Kendal et al., 2018; Schultner et al., 2024). Together, these findings indicate that agents can build upon fragmented experiences to gradually refine their world models over generations, mirroring learning dynamics observed in human populations (Tessler et al., 2021).

## 6 DISCUSSION

Our approach aligns with a longstanding research program in which human mental representations are modeled as structured, program-like generative theories, and learning is understood as probabilistic inference over these structures (Griffiths et al., 2010; Lake et al., 2017; Rule et al., 2020). This perspective has been highly successful in explaining human causal reasoning (Tenenbaum et al., 2006; Griffiths and Tenenbaum, 2009), intuitive physics (Battaglia et al., 2012; Smith et al., 2023), or social reasoning (Baker et al., 2011; Ying et al., 2024): people form hypotheses about latent mechanisms,

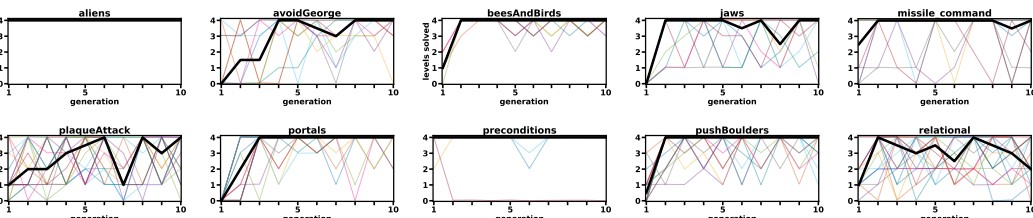

Figure 6: **Generational learning via linguistic advice.** Median performance of $N = 20$ agents (black) across generations (2 lives each). Our models affords generational learning: performance increases across generations, beyond the performance of individual agents (generation 1).

simulate their consequences, and revise them in light of new evidence. Executable, program-like world models are a natural next step in this line of work (Tsividis et al., 2021), and our contribution can be seen as extending this framework to the domain of social learning by integrating linguistic guidance and direct experience within a unified inferential model.

Our experiments showed that our computational model reproduces key features of human social learning: advice reduces the attempts needed for success by shaping exploration, supports generational knowledge accumulation, and even allows model-generated guidance to help human learners—demonstrating bidirectional knowledge exchange.

While VGDL is only a coarse approximation of human game representations, our results show it captures enough structure to study social learning and afford bidirectional human–model knowledge transfer. Future work could leverage library-learning methods (Ellis et al., 2020; Wong et al., 2021) to model the emergence of shared representations through linguistic interaction, potentially driving representational convergence rather than requiring pre-aligned representations.

Our results show that linguistic guidance often speeds and safeguards exploration, but when advice is wrong it can restrict search — mirroring the "double-edged sword of pedagogy" observed in developmental psychology (Bonawitz et al., 2011). Humans counter this issue by evaluating testimony against prior causal theories before integrating it (Harris et al., 2018; Sobel and Kushnir, 2013). To match this sophistication, computational models will need meta-cognitive mechanisms to judge the reliability of advice and adjust their exploration accordingly.

This paper opens several avenues for future work. Human-generated messages often include game abstractions, high-level strategies, and planning heuristics that our model currently cannot leverage. Extending the framework to interpret and learn from these richer forms of guidance — e.g. through inference of auxiliary reward functions, planning abstractions and strategies — could unlock enhanced social learning capabilities (Silver et al., 2024, e.g., ). It would also be valuable to examine in more detail which linguistic abstractions — beyond rule and win/loss information — facilitate robust generational transfer, building on methodologies from prior human iterated-learning studies in VGDL environments (Tessler et al., 2021). Beyond passive learning, our model could be further extended to make decisions about who to learn from based on perceived expertise or success — a capacity known as prestige-based social learning (Kendal et al., 2018; Schultner et al., 2024). Future work could also investigate how different LLM families behave within our framework in their dual roles as advice generators (speakers) and approximations of human speakers (speaker models). Because our method uses LLMs both to produce pedagogical messages and to evaluate the likelihood of human-written advice, comparing families with different inductive biases could reveal how model-specific language priors shape both message interpretation and learning outcomes.

From an artificial intelligence perspective, an important direction for future work is extending this framework to real-world, continuous, and more complex environments. Doing so will require advances in program synthesis, scalable probabilistic inference, and hardware capable of performing inference over rich, unstructured programming languages such as Python (Tang et al., 2024; Lehrach et al., 2025). This is an active and rapidly growing area of research,[2] with significant investment and recent progress in LLM-driven code generation and executable world modeling (Cusumano-Towner et al., 2019; Lew et al., 2023; Loula et al., 2025). Although this may seem challenging, humans routinely construct rich executable models in code — physical engines, video game environments, simulation of complex systems — which allow them to reason about highly complex processes, explore counterfactual scenarios, and deepen their understanding of the world. These practices illustrate the feasibility and potential benefits of scaling program-like world models to richer domains.

Finally, our results point to exciting possibilities for human-machine collaborative learning. Our model not only benefits from human-generated guidance but also contributes back through effective, pedagogical advice — closing the collaborative loop. This demonstrates a first step toward bidirectional learning systems capable of supporting human learners. The future directions outlined above — handling richer linguistic guidance, adaptive trust in information sources, and scaling to open-ended domains — would represent major steps toward AI systems that not only learn efficiently but also teach, collaborate, and adapt within complex social learning networks, augmenting collective intelligence in hybrid human–AI communities (Colas et al., 2022; Brinkmann et al., 2023; Collins et al., 2024).

---

[2]e.g., see recent library scaling probabilistic programming with GPUs via Jax.

## REPRODUCIBILITY STATEMENT

Section 3 details the full model specification and inference procedure. Section 4 describes the human and model experimental design. The appendices provide additional details including: the description of all VGDL primitives, the inference pseudo-code, details about guided proposals and planning, instructions used for human data collection and full prompts used by the LM components of our model. The codebase will be released at github.com/ccolas/language_and_experience. Together these resources allow full replication of the experiments.

## ACKNOWLEDGEMENTS

Cédric Colas is partly funded by the European Union's Horizon 2020 research and innovation programme under the Marie Skłodowska-Curie grant agreement No 101065949. This project was supported by a Intel and the National Science Foundation under grants CCF-2217064 and IIS-2212310. Research was additionally sponsored by the Department of the Air Force Artificial Intelligence Accelerator and was accomplished under Cooperative Agreement Number FA8750-19-2-1000. The views and conclusions contained in this document are those of the authors and should not be interpreted as representing the official policies, either expressed or implied, of the Department of the Air Force or the U.S. Government. The U.S. Government is authorized to reproduce and distribute reprints for Government purposes notwithstanding any copyright notation herein.

## AUTHOR CONTRIBUTIONS

CC led the development of the computational model, conducted simulation experiments, and wrote the initial draft of the paper. TM and BP led the design and implementation of the human experiments, including participant recruitment and data collection. CC, MHT, NG, JA, and JT contributed to the formulation of research questions, the high-level design of the computational framework, and editing the manuscript. All authors provided feedback and helped shape the final version of the paper.

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

# Appendices

APPENDIX CONTENTS

## A  EXTENDED RELATED WORK

**Language and RL.** Language has emerged as a powerful tool to enhance reinforcement learning (RL) by conveying state abstractions (Narasimhan et al., 2018), world dynamics (Zhong et al., 2020), auxiliary reward functions (Goyal et al., 2019), and task decompositions (Shridhar et al., 2021; Sharma et al., 2021). Its inherent compositionality and abstraction capabilities allow agents to pursue more abstract goals (Jiang et al., 2019), generalize effectively across diverse environments (Zhong et al., 2020) and goals (Hill et al., 2020; Colas et al., 2020), and structure long-term decision-making more effectively (Hu et al., 2019; Chen et al., 2021). Despite these advances, most language-based RL approaches rely on substantial amounts of paired data to ground linguistic information in agent experience, limiting their scalability and resemblance to human-like social learning. Language models (LMs) promise more flexible combination of language and decision-making (Ahn et al., 2022; Huang et al., 2022), yet they struggle to learn complex embodied skills that require low-level perception and temporally extended actions (Valmeekam et al., 2023; Paglieri et al., 2024). Nottingham et al. (2023) use LLMs to generate complete world model hypotheses that are then verified through experience. Our approach introduces a Bayesian framework that treats experience and language as two complementary sources of evidence in the inference of world models. This joint inference enables rapid adaptation to new linguistic inputs and tasks from the very first interaction, bypassing the need for extensive paired data.

**Bayesian models of social cognition.** Bayesian models of social cognition have proven to be powerful tools for modeling theory of mind (ToM)—the ability to infer the hidden goals, beliefs, and intentions of others from observable behavior or linguistic cues (Baker et al., 2011; 2017; Frank and Goodman, 2012; Goodman and Frank, 2016). These models formalize social inference as an inverse planning problem, where observers assume that agents act approximately rationally towards their

goals and use this assumption to infer likely mental states (Baker et al., 2011; 2017). Extensions of these frameworks to language understanding have resulted in the *Rational Speech Acts* model, which interprets communication as recursive social reasoning: listeners reason about what a speaker intends to convey based on the assumption that speakers choose utterances optimally, given their own beliefs (Frank and Goodman, 2012; Goodman and Frank, 2016). More recent work integrates Bayesian reasoning with modern machine learning to enable richer social inferences. For instance, (Zhi-Xuan et al., 2023) leverage large language models (LLMs) as priors in a Bayesian goal inference system, allowing for the efficient suggestion and evaluation of likely goals in complex environments. Similarly, (Ying et al., 2024) introduce a language-augmented ToM model that translates natural language statements about beliefs into formal epistemic representations, enhancing agents' ability to reason about others' knowledge and intentions. By combining linguistic input with structured probabilistic models, these approaches extend traditional ToM beyond purely behavioral cues, opening new avenues for interactive and socially aware AI agents (Vélez and Gweon, 2021). However, these models operate in settings where the transition dynamics are fully known and deterministic, and inference is restricted to identifying a latent goal or belief state from demonstrations. In contrast, our setting requires agents to infer the entire causal structure of each new environment — object types, interaction rules, and win/loss conditions—directly from experience, making joint inference from language and action substantially more challenging. Our work builds on these ideas by leveraging Bayesian inference to jointly interpret linguistic and experiential data, allowing agents to efficiently acquire world models from sparse social interactions.

# B    LEARNING FROM EXPERIENCE ALONE

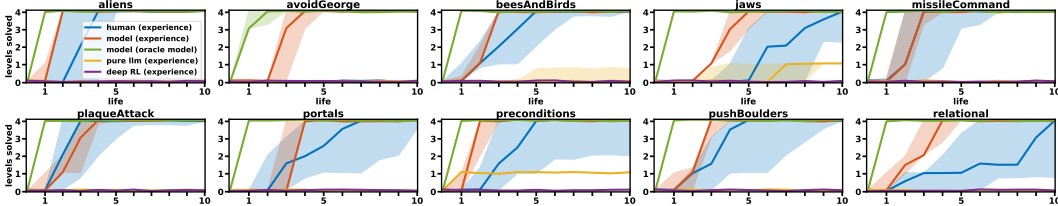

Figure 7: **Learning from experience.** Both humans and our model achieve high sample efficiency across most games. Systematic differences emerge in *relational*, which requires systematic testing of object combinations, and *avoidGeorge*, which demands rapid and precise planning. *Deep RL* and *pure LM* baselines fail to solve games efficiently, highlighting the value of structured representations (N=20, median ± interquartile range).

# C    VGDL: GAME PRIMITIVES, STATE AND ACTION SPACES

We work with a subset of the VGDL domain. Here are the possible types of avatars:

- *MovingAvatar*: controllable player that can move in the four directions with a certain `speed` based on keyboard presses
- *ShootAvatar*: MovingAvatar that can also shoot objects `stype` when the player presses the space bar
- *FlakAvatar*: ShootAvatar that can only move sideways and always shoot upwards.

Here are the possible object types and their `parameters`:

- *Immovable*: object that cannot move
- *Flicker*: object that disappears after `total` steps
- *SpawnPoint*: object that spawn objects `stype` with probability `p`
- *ResourcePack*: object that can be collected (see interaction *addResource* and *removeResource*)
- *Passive*: object that can be pushed (see interaction *bounceForward*)
- *Missile*: object that moves in one direction with a certain `speed` and an original `orientation`. They can change direction (see interactions *turnAround* and *reverseDirection*

- *Bomber*: the combination of a missile and a spawner (with their combined parameters)
- *Chaser*: object that moves in the direction of the nearest target object `stype`, with a certain *speed*
- *RandomNPC*: object that moves in a random direction with a certain `speed`
- *Portal*: object that can teleport another object contacting it to another exit object (see interaction *teleportTo*).

All moving objects move every `cooldown` environment steps.

Interactions describe what happens when two objects contact. Here are the possible interaction types and their `parameters`:

- *noInteraction*: nothing happens
- *killSprite*: the second object kills the first object
- *transformTo*: the second object transforms the first object into a third object `stype`
- *removeResource*: the second object decreases the count of `resource` from the first object
- *addResource*: the second object increases the count of resource of the first object
- *killIfHasLess*: the second object kills the first if it has less than 1 resource `stype`
- *stepBack*: objects step back (second steps back first, if not possible the second does)
- *bounceForward*: the second object pushes the first if possible (e.g., unless it is blocked)
- *turnAround*: the first object (a missile) does one block down and switches direction when encountering the second object
- *reverseDirection*: the first object (a missile) reverses direction when encountering the second object

killSprite and transformTo interactions can further lead to a positive (+1), negative (-1) or null (0) reward.

Lose conditions can be:

- *Timeout*: the player loses if it runs out of time before solving the task
- *CountIsZero*(objs): the player loses if at least one of the objects `objs` has no remaining instances in the game.

Win conditions can be:

- *Survive*: the player wins if it survives long enough
- *CountIsZero*(objs): the player wins if the count of all objects `objs` goes to zero (e.g., they were all killed or disappeared)

**Size of the search space and game difficulty.**

The size of the search space is a direct function of the number of objects. The avatar must have an *avatar type* among 3, each other object should have an *object type* among 10; each pair of objects must have an interaction type among 10; win and lose conditions can apply to any list of objects from 0 to all. Without accounting for type parameters (e.g., chasing object have a target, transform interaction transform objects into a specific type, etc), game spaces already scale $10^{30}$ for the smallest games involving 5 objects (*missileCommand*, *preconditions*). Other games might have up to 10 objects (*plaqueAttack*, *portals*). The difficulty of a game is not necessarily correlated to the size of the search space. For instance, some object types can be inferred quickly from movement observations, while others require direct interactions. The layout and game rules might also make exploration, or planning more or less complex for different games.

**Perception and action.** Players have five possible actions: four directional moves (arrow keys) and a shoot action (space bar) when the game includes a *ShootAvatar* or a *FlakAvatar*. Computational agents also have access to a NOOP action, while human players can simply choose not to act. After taking a non-NOOP action, computational agents must wait for 4 environment steps before being able to take a new one, a way of capturing human reactive time (5 fps). The game runs at 20 frames per second. Humans perceive the game visually, with the current reward displayed at the top (see

Figure 2), whereas computational agents perceive symbolic states consisting of object properties (object id, color, position), rewards, and win/loss events.

# D  INFERENCE PSEUDO-CODE

Algorithm 1 shows the pseudo-code of the inference algorithm — a particle filter with Metropolis rejuvenations — used to update the agent's beliefs about possible games it might be playing, supported by experiential and linguistic evidence (Del Moral et al., 2006; Metropolis et al., 1953). The *Perturb* operator replaces one rule of the VGDL game description at a time, according to guided proposals (see Appendix Section E below). We picked the number of particles to be $N = 20$. This involves a trade-off:

- Exploration: too few theories lead to under-representation of uncertainty, reducing the model's ability to identify informative subgoals and to explore sufficient alternative variations in the space of possible theories.
- Computation: each Metropolis update requires updating all particles and computing language likelihoods via the LLM, which is the computational bottleneck of the approach.

Empirically, 10 particles was faster but sometimes failed to discover high-posterior theories quickly enough for the agent to survive, while 20 ensured a more robust and fast exploration of the theory space. More particles might slightly improve the performance of the inference, but will make the system slower to run and experiment with.

---

**Algorithm 1** Particle filter with Metropolis rejuvenation

---

**Params:** $N$ number of particles/theories, $p(T)$ prior over theories, $S, R$ number of steps and rejuvenation steps.
**Inputs:** inference step $s$, new experiential data $D_s$, particles $\boldsymbol{\theta_s}$ (for $s > 0$).
$\qquad\qquad\qquad\qquad\qquad\qquad\qquad\qquad\qquad\qquad$ ▷ **bold** indicates vectors of size $N$
**if** $s == 0$ **then** $\qquad\qquad\qquad\qquad\qquad\qquad\qquad$ ▷ Sample initial particles from prior
$\quad$ $\boldsymbol{T_0} \sim p(T)$
**for** $s \in [1, S]$ **do** $\qquad\qquad\qquad\qquad\qquad\qquad\qquad\qquad\qquad\qquad$ ▷ inference step
$\quad$ $\boldsymbol{T_s} \leftarrow \boldsymbol{T_{s\text{-}1}}$
$\quad$ **for** $r \in [1, R]$ **do** $\qquad\qquad\qquad\qquad\qquad\qquad\qquad$ ▷ MCMC rejuvenation step
$\qquad$ $\boldsymbol{T_s'} \leftarrow \text{Perturb}(\boldsymbol{T_s})$ $\qquad\qquad\qquad\qquad\qquad\qquad\qquad$ ▷ Propose move
$\qquad$ $\boldsymbol{\alpha} \leftarrow \frac{p(\boldsymbol{D_s}, \boldsymbol{T_s'})}{p(\boldsymbol{D_s}, \boldsymbol{T_s})}$ $\qquad\qquad\qquad\qquad\qquad$ ▷ Metropolis acceptance
$\qquad$ **for** $i \in [1, N]$ **if** $u \sim U(0, 1) < \alpha^i$ **do**
$\qquad\qquad$ $T_s^i \leftarrow T_s^{'i}$ $\qquad\qquad\qquad\qquad\qquad\qquad\qquad\qquad$ ▷ Accept move
$\quad$ $\boldsymbol{w} \leftarrow \text{Norm}(\boldsymbol{p}(\boldsymbol{D_s}, \boldsymbol{T_s}))$ $\qquad\qquad\qquad\qquad\qquad$ ▷ Update weights
$\quad$ $\boldsymbol{T_s} \leftarrow \text{Resample}(\boldsymbol{T_s}, \boldsymbol{w})$ $\qquad\qquad\qquad\qquad$ ▷ SMC resampling step
**Return:** $arg\,max_{T_{t+S}^i \in \boldsymbol{T_{t+S}}}\ p(D_s, T_{t+S}^i)$ $\qquad\qquad$ ▷ Return MAP theory

---

# E  GUIDED PROPOSALS

We use biased proposals to initialize the set of 20 candidate theories and to generate rejuvenation moves (*Perturb* operator in Algorithm 1). These proposals are guided by both experience and linguistic evidence, accelerating convergence towards theories that better explain the agent's observations.

Experience-driven proposals bias the sampling process in the following ways:

- objects observed to have moved cannot be assigned an object type incompatible with movements (e.g., *Immovable*, or *Flicker*),
- objects moving in one direction are more likely to be assigned type object types that move linearly (*Missile* and *Bomber* than object types that allow movements in all directions (*RandomNPC*, *Chaser*),
- objects pairs involved in collisions preceding observed rewards are more likely to be assigned reward-generating interactions.

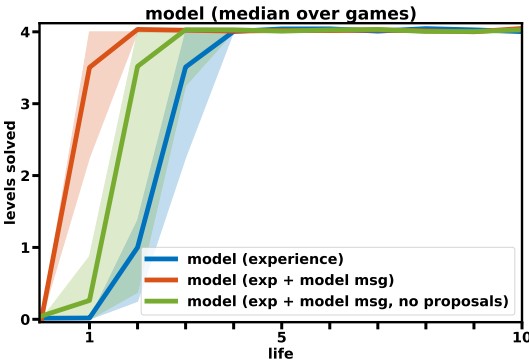

Figure 8: **Ablation of language-guided proposals.** a) Median performance across games for models learning from experience alone (blue), or experience and language, using language likelihood and language-guided proposals (orange), or ablating language-guided proposals (green). (N=20, error bar: interquartile range).

## F    DETAILS ABOUT PLANNING

Our model strategically balances curiosity-driven exploration with goal-directed exploitation, mirroring human problem-solving strategies (Tsividis et al., 2021). From the current best theory $T_{\text{MAP}}$, we generate candidate subgoals: specific collisions between pairs of objects that the agent can cause to occur by either touching another object itself, or by pushing or spawning an object onto it. Candidate subgoals are assigned values based on their exploration and exploitation potentials:

$$\text{Value}(g) = \text{ExplorationValue}(g) + \text{ExploitationValue}(g).$$

The exploration value measures disagreement between theories in the current population:

$$\text{ExplorationValue}(g) = 1 - \frac{\max_i \text{count}(i \mid g)}{\sum_i \text{count}(i \mid g)}$$

where $\text{count}(i \mid g)$ is the count of theories assigning interaction type $i$ to subgoal $g$.

The exploitation value rewards key game mechanics:

$$\text{ExploitationValue}(g) = \begin{cases} 10 & \text{if } g \text{ contributes to win condition} \\ 8 & \text{if } g \text{ protects essential resources} \\ 6 & \text{if } g \text{ creates useful tools} \\ 2 & \text{if } g \text{ collects resources or rewards} \\ 0 & \text{else} \end{cases}$$

To achieve these goals, the model evolves short 10-step action sequences (e.g. *move left, shoot, move up*) through stochastic search, using $T_{\text{MAP}}$ to simulate outcomes. Action plans are iteratively mutated and refined over three generations using a simple genetic algorithm, where mutations crop and regrow action sequences from a uniformly sampled mid-point. Each sequence $a$ is evaluated according to:

$$V(a) = R_{\text{game}}(a) + R_{\text{goal}}(a) + R_{\text{win/loss}}(a),$$

where $R_{\text{game}}$ is the game's reward function under $T_{\text{MAP}}$, $R_{\text{goal}}$ rewards progress toward the selected subgoal (between 0 and 1), and $R_{\text{win/loss}}$ provides +100 for winning and -100 for losing.

Additionally, the model performs 10 3-step lookaheads to avoid catastrophic errors, triggering replanning when the originally predicted value deviates significantly from the distribution of values for newly simulated trajectories:

$$V(a_{\text{original}}) > \max_{i \in 1\ldots10} V(a_{\text{lookahead}_i}).$$

This safety mechanism prevents the agent from executing plans that appeared promising under limited simulation but fail under more extensive testing, or when unexpected changes in the environment render the original plan ineffective.

## G  HUMAN DATA COLLECTION: INSTRUCTIONS AND PARTICIPANT RECRUITMENT

We recruited 122 adult English-speaking participants through Prolific to play 5 randomly-assigned games. To ensure task engagement while maintaining a representative sample, we excluded participants who failed to complete at least one level in $\geq 3$ games (final N=120). Participants' median completion time was 49.30 minutes, and the median hourly pay rate was $10.41/hr.

Before playing the games, participants read the following instructions:

---

**Instructions for human data collection**

```
In this experiment, you will play different games before describing
each game to another participant who has not yet played.
You will play 5 different games. Each game will be played using the
arrow keys and spacebar on your keyboard.
You will start each game with 15 lives. Each game has 4 levels. Your
goal is to win all 4 levels of the game in as few lives as possible.

You lose a life when you lose a level. There are different ways of
winning and losing. Possible ways to win a level include removing all
 objects of particular colors, reaching a particular object, or
surviving for long enough.

Possible ways to lose a level include allowing all objects of
particular colors to die or disappear, or by not solving the level
fast enough.

You move on to the next level of a game only after winning the
current level. You will finish a game when you win all 4 levels, or
when you have lost 15 lives.

After finishing a game, you will be asked to describe the game to
another participant who has not yet played. Your goal is to help them
 solve the game in as few lives as possible. You will be required to
spend at least 30 seconds describing each game.

\textbf{Bonus opportunity: Solve each game in as few lives as
possible!}
On each game, you can win up to $\$0.20$ based on your performance.
Your rank on each game will be computed relative to other
participants, based on the number of levels you complete, and how
many lives you take to complete them. For each game you will earn the
 max bonus multiplied by the proportion of other participants you
outrank!

\textbf{Bonus opportunity: Write useful descriptions!}
On each game, you can win up to an additional $\$0.20$ based on your
description. Another group of participants will read descriptions
before playing each game, and receive bonuses up to $\$0.20$ based on
 their relative rank. On each game, you will additionally receive the
 bonus earned by the player who reads your description, so help them
win!

// [Optionally, for participants in condition 2 and 3 (social
learning)]
When starting each game, you will be shown a message from another
player who has already played the game. You should use this message
to help you play the game, but keep in mind that it's possible that
it contains mistakes.
```

---

When starting each game, participants in Condition 1 received no game specific advice, as shown in Figure 9a, while participants in Conditions 2 and 3 saw a message from a previous player, as shown

in Figure 9b. Players in Conditions 2 and 3 can still read the message as they play. After completing each game, participants in all conditions were asked to describe the game to a future player, as shown in Figure 9c.

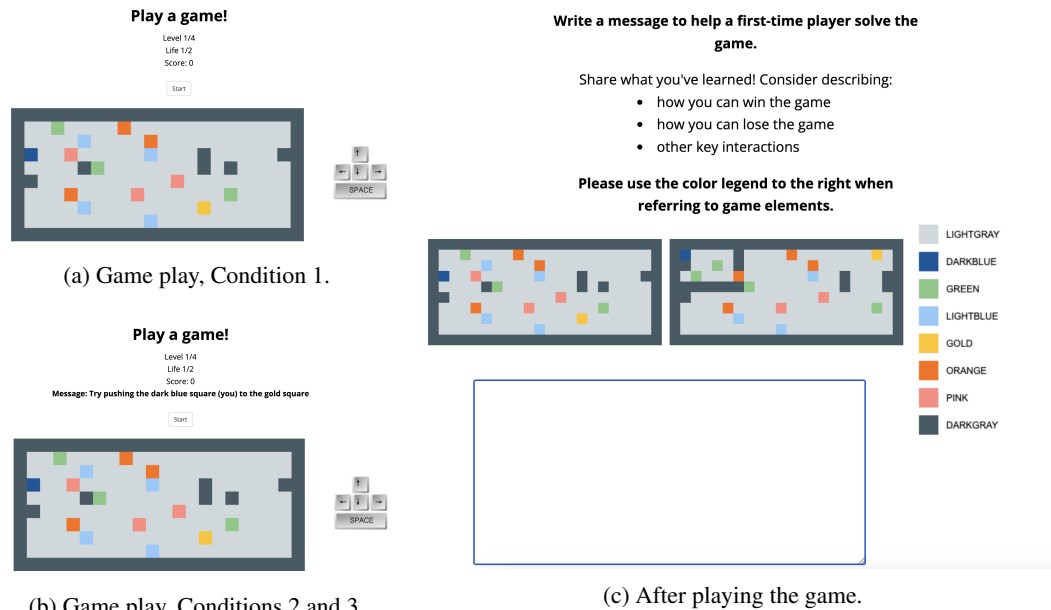

(a) Game play, Condition 1.

(b) Game play, Conditions 2 and 3.

(c) After playing the game.

Figure 9: Screenshots of the game screens for the different experimental conditions. (a) Condition 1: individual gameplay instructions. (b) Conditions 2 and 3: social gameplay instructions. (c) Message interface shown to participants for writing advice to future players. Note that players in the social conditions can see the message during gameplay.

## H    COMPUTATIONAL RESOURCES

The model requires a GPU to run the LM, we use one NVIDIA A100 (80Gb). We use prompt caching using the vLLM library to speed up the generation of proposals. Simulation runs vary in function of game complexity (slower with more objects) and the learning speed of the agent (runs end early when all levels are solved), taking anywhere between an hour and a day, depending on games and conditions.

## I    EXAMPLE MESSAGES

Below are example messages generated by humans and model players in the social learning experiments.

---

**Example of human messages in social learning experiment**

```
[beesAndBirds] "Move the darkblue square by pressing the arrows keys.
 You win the game the touching the green square, you can also touch
red but not needed. Avoid touching the moving yellow square because
they can kill you. The green squares can destroy the yellow and
orange squares when you free it from the purple squares. You can
destroy purple squares by touching it."

[aliens] "You must use space bar to shoot and hit the pink moving
squares. You are also needed to dodge the red squares that are being
shot by the pink squares. You will dodge by using the left, right, up
```

and down keys. You must destroy all the pink squares before they
reach the bottom."

[jaws] "I think if you move the block over the purple and reds you
will win the game. You will lose the game if you go over other color
blocks."

[preconditions] "eat the white tiles to enable you knock down the
green tiles toget to the yellow tiles. your objective is to get to
the yellow tiles"

[missile_command] "So the main goal is to protect the GREEN tiles by
making sure the RED tiles and the YELLOW tiles don't touch them and
you can win even if you only have ONE GREEN tile left. As soon as you
 click START, they will start moving and you need to act fast by
trying to interact with the RED and YELLOW tiles to make them
disappear using the SPACEBAR while next to them. The YELLOW tiles
move faster than the red so try to prioritize them if possible but
there will be times that you would need to sacrifice one GREEN tile
to make one YELLOW tile disappear because at that same time, you can
possibly eliminate 2-3 RED tiles because they are slower. You win the
 level by keeping at least one GREEN tile intact and you lose when
you lose all five of them."

[pushBoulders] "Move darkblue square by using the arrow keys. Avoid
touching the orange squares because they can kill you, push the green
 into orange to make orange disappear. Avoid pink square because it
can kill, move pink by pushing green into it. You can touch lightblue
 square to make it disappear. Also you can touch yellow square to
make it disappear and win the game."

[avoidGeorge] "Game 4: Move your dark blue square quickly to avoid
the light blue square. Use the brown spacebar to turn the green
squares back to purple. Shoot the squares next to you with the brown
square to eliminate them."

[plaqueAttack] "Use your arrows to shoot from your square to protect
the red and yellow squares. If the other colors get to red and yellow
 and turn them before you can eliminate them, you lose."

[relational] "Use the arrow keys to move the orange cubes to merge
the darkblue and white cubes by pressing the space bar key. Avoid
colliding with the rest of the cubes in the game"

[portals] "You need to move your character over the gold and light
blue squares. To win the game you need to get to the light green
square. In the levels coming up you needs to move up from light blue
to dark blue to get to different areas. Do not let the other colors
touch you because you will lose a life."

---

**Example of model messages in social learning experiment**

[beesAndBirds] "Use arrow keys to move your darkblue. To win,
eliminate all green by touching them, earning points for each kill.
Be careful: you'll die if you touch yellow or orange, but you can use
 lightgreen to your advantage - it can kill both yellow and orange.
Touch red and purple to kill them, but they're not your main
objective."

[aliens] "Use left and right arrow keys to move your darkblue square
and spacebar to shoot blue squares. To win, eliminate all orange,
pink, and brown - shooting blue at pink will kill them and earn

points, but watch out as pink can kill you on contact. Protect
yourself from pink and red, which can also kill you, and avoid
touching them. Blue objects can help by killing pink, but be aware
that blue and white objects kill each other, and pink can generate
more red objects."

[jaws] "Control your darkblue square with arrow keys. Your goal is to
 survive for as long as possible. Be careful, as orange, brown, and
pink objects will kill you on contact – orange objects will even
chase you. Stay away from them and try to navigate around the safe
darkgray, purple, and green objects to stay alive."

[preconditions] "Use arrow keys to move your darkblue square. Your
goal is to eliminate all gold objects by touching them to earn points.
 Be careful, as green objects will kill you on contact unless you
have a white resource to protect yourself – collect white resources
to stay safe. You can also kill green objects by touching them, but
they'll take a resource from you in the process."

[missile_command] "Use arrow keys to move the darkblue square and
spacebar to shoot pink. To win, eliminate all red and gold – shoot
pink at them to kill and earn points. Protect green objects, as you'
ll lose if they all die, and try to avoid letting red and gold kill
them as this will also cost you points. Red and gold objects will
chase green, so use this to your advantage and take them out with
pink squares."

[pushBoulders] "Use arrow keys to move your darkblue square. To win,
eliminate all gold objects by touching them or pushing green onto
them – you'll earn points for each gold destroyed. Be careful:
touching orange or pink will kill you. Green is a valuable ally, as
it can also kill gold and earn you points, so use it to your
advantage by pushing it onto gold. Protect yourself and keep pushing
green to clear the way to victory, as dying means losing the game."

[avoidGeorge] "Use arrow keys to control your darkblue square and
spacebar to shoot brown. To win, survive long enough while keeping
green objects alive – they can be created by shooting brown at purple
 or letting orange touch purple. Watch out: lightblue objects will
kill you on contact and can also turn green back into purple, so
protect your green! You'll earn points when brown or orange transform
 purple into green, but lose points if lightblue kills you."

[plaqueAttack] "Use arrow keys to move your darkblue and spacebar to
shoot lightred. Your goal is to eliminate all purple, orange, brown,
and lightblue while keeping yellow and red alive. To help you,
lightred objects will kill orange and brown on contact, so position
them to clash. You can also shoot lightred at orange and brown to
kill them directly. Touching green objects transforms them into
yellow, earning you points, and touching blue objects transforms them
 into red, also earning points. Be careful, as orange and brown can
transform yellow into green, losing you points, and blue objects can"

[relational] "Use arrow keys to move your darkblue square. To win,
eliminate all blue objects, which can be done by pushing them onto
yellow objects – when they kill each other, you gain points. Be
careful not to touch white objects, as they'll kill you. You can also
 transform objects to help your strategy: touch red to turn them into
 yellow, touch pink to turn them into orange, and push orange onto
purple to turn them into yellow."

[portals] "Use arrow keys to move your darkblue square. Your goal is
to eliminate all lightgreen objects by touching them to earn points
and win the game. Be careful – darkgreen, pink, and lightred objects

```
will kill you on contact, so avoid them. You can teleport using
lightblue to gold and blue to lightorange to help navigate. Brown
objects move randomly, but won't harm you, and other objects are
stationary, so use them to your advantage."
```

## J PROMPTS

**Prompt for language likelihood estimation and language generation**

```
# Game Message Generation Task

You are writing standardized messages to describe a 2D video game
based on the game's mechanics.

### Instructions
Each message should explicitly state:
- How to control the game
- How to win
- How you could lose/die
- How to get/lose points
- Key game mechanics that aren't obvious while playing

### Context
- The player controls the darkblue object
- In some games, it can shoot other objects of a particular color
- Each object color is associated to a particular object type
conditioning how they move
- Effects can apply when two colors collide, eg one can kill the
other, one can transform the other into a third color, one can push
the other, etc
- Possible wining conditions include: surviving long enough, or
killing all objects of one or several target colors
- Possible losing conditions include: reaching a timeout before
wining, or seeing all objects in a list of colors dying/disappearing

### Writing Rules
- Mention all important interactions in the game
- State each interaction in a single clear sentence
- List all killing and transformation interactions
- Always specify win and loss conditions

### Example

Who you are and how you move:
- You control the darkblue square with arrow keys.
How you win and lose:
- You get points when pink objects transform blue objects into purple
 objects.
- You lose points when darkred objects kill purple objects.
- You win the game when all blue and orange objects are dead.
- You lose if all purple and lightred objects die.
What you can do:
- You can shoot pink squares by pressing space bar.
- You can kill orange objects by shooting pink objects at them.
- You can transform blue objects into purple objects by shooting pink
 objects at them.
What can kill you:
- Orange objects will kill you if you touch them.
Other possible interactions:
- You can push lightred objects.
```

```
- Darkred objects can kill purple objects by touching them.
- Darkred objects can kill lightred objects by touching them.
Other objects:
- Pink objects move along one direction.
- Darkred objects chase the nearest purple object.
- Orange objects move along one direction.
- Purple and blue objects do not move.

Message:
You control a darkblue square with arrow keys. You can shoot pink
objects by pressing space bar. Orange objects kill you when they
touch you. Pink objects kill orange objects when they touch them.
Pink objects transform blue objects into purple objects when they
touch them. You can push lightred objects. Darkred objects kill
purple and lightred objects when they touch them. Pink objects move
in one direction. Darkred objects chase the nearest purple object.
Orange objects move in one direction. Purple and blue objects don't
move. You get points when pink objects transform blue objects into
purple objects. You lose points when darkred objects kill purple
objects. You win when all blue and orange objects are dead. You lose
if all purple and lightred objects die.

### Task

{THEORY DESCRIPTION}

Message:
{MESSAGE} # we compute the likelihood of these tokens
```

---

**Prompt for making rules proposals**

```
# Game Rules Analysis Task

You are analyzing messages written by video game players to help you
play a game for the first time. These games feature colored objects
interacting in a 2D space, where players control a darkblue object.
The messages describe controls, scoring systems, win/loss conditions,
 hazards, core mechanics, and give you tips to help you win the game.

### Your Task
You will be asked to answer four types of questions:
- What is the type of objects of [color1]?
- What happens to objects of [color1] when they collide with objects
of [color2]?
- Does the player need to kill all objects of [color1] to win?
- Does the player lose when all objects of [color1] die?

Answer questions by looking for explicit statements in the message
about:
- How objects move and how they behave, eg can you use them as a
portal, can you push them, do they chase other objects, etc.
- Objects killing other objects
- Objects transforming into other objects
- Win conditions involving killing objects
- Loss conditions involving object death

### Rules for Answering
- Only use information explicitly stated in the message
- You are the darkblue object.
- Do not make assumptions about interactions not mentioned
```

– Pick the "I don't know" answer if the corresponding information is ambiguous, contradicting, or absent
– Pay attention to interaction direction – if A kills B, it doesn't mean B kills A
– Each question comes with a fixed set of possible answer. You must pick one of the answer. If the correct answer is not in the list, pick the "I don't know" answer.
– If the way to win is to kill some object, then having all these objects disappear/die does NOT cause you to lose
– If you need to "protect [color1] objects" or "kill [color2] objects before they kill / transform [color1] objects", you must answer:
  – "You lose if all [color1] objects die or disappear", note that transformations kill the transformed objects.
– If you need to reach / touch / kill by touching [color1] objects to win, this means you can kill color1 objects. You MUST answer:
  – "[color1] objects die when they collide with darkblue objects".
– If the message says you can "collect [color1]":
  – either [color1] is a resource, and you must answer that "you can collect [color1] resources"
  – OR [color1] is not a resource, and you must answer that "you can kill [color1] objects"
– If the message says you can destroy/kill/eliminate [color1] objects if you have enough [color2] resources, the two following statements are true:
  – "[color1] objects die when they collide with darkblue objects but also take a [color2] resource from them'
  – AND "darkblue objects die if they don't have enough resources when they collide with [color1] objects"
– Objects that kill you when you lack resources also die when they touch you, and take one resource from you.
  – "[color1] objects die when they collide with darkblue objects but also take a [color2] resource from them"
– When [color1] objects take away some resource / your health when you touch them, they also die:
  – "[color1] objects die when they collide with darkblue objects but also take a [color2] resource from them'
– Use the ... after "Reasoning:" to reason about the game before generating your answer.

### Example Game 1

Message from the player:
"Orange objects kill you when they touch you. You kill purple objects when you touch them. Lightgreen objects chase orange objects and can kill them when they touch them. You win when you kill all purple objects. You lose if orange kills you."

--

What happens to darkblue objects when they collide with orange objects?
1) nothing happens to darkblue objects when they collide with orange objects
2) darkblue objects die when they collide with orange objects
3) darkblue objects get transformed when they collide with orange objects
4) darkblue objects steal resource from orange objects when both collide
5) darkblue objects die if they don't have enough resources when they collide with orange objects
6) I don't know / something else

Reasoning: ......................

```
Answer (pick one from the list):
2) darkblue objects die when they collide with orange objects

--

What happens to orange objects when they collide with darkblue
objects?
1) nothing happens to orange objects when they collide with darkblue
objects
2) orange objects die when they collide with darkblue objects
3) orange objects get transformed when they collide with darkblue
objects
4) orange objects steal resource from darkblue objects when both
collide
5) orange objects die if they don't have enough resources when they
collide with darkblue objects
6) I don't know / something else

Reasoning: ......................

Answer (pick one from the list):
6) I don't know / something else

Note: orange kill darkblue (the player), but we don't know what
happens to orange when they touch (maybe nothing)

--

How do orange objects behave?
1) orange objects cannot move
2) orange objects do not move and disappear after a certain time
3) orange objects regularly spawn/generate other objects
4) orange objects can be pushed
5) orange objects move along one axis
6) orange objects move along one axis and regularly spawn/generate
objects
7) orange objects chase or flee another object
8) orange objects move randomly
9) orange objects are portals
10) I don't know

Reasoning: ......................

Answer (pick one from the list):
10) I don't know

--

How do lightgreen objects behave?
1) lightgreen objects cannot move
2) lightgreen objects do not move and disappear after a certain time
3) lightgreen objects regularly spawn/generate other objects
4) lightgreen objects can be pushed
5) lightgreen objects move along one axis
6) lightgreen objects move along one axis and regularly spawn/
generate objects
7) lightgreen objects chase or flee another object
8) lightgreen objects move randomly
9) lightgreen objects are portals
10) I don't know

Reasoning: ......................
```

```
Answer (pick one from the list):
7) lightgreen objects chase or flee another object

--

Do you need to kill purple objects to win?
1) To win you need to reach/touch purple objects
2) To win you need to kill all purple objects
3) To win you don't need to kill all purple objects
4) To win I don't know if you need to kill or reach/touch all purple
objects

Reasoning: ......................

Answer (pick one from the list):
2) To win you need to kill all purple objects

--

Do you need to kill orange objects to win?
1) To win you need to reach/touch orange objects
2) To win you need to kill all orange objects
3) To win you don't need to kill all orange objects
4) To win I don't know if you need to kill or reach/touch all orange
objects

Reasoning: ......................

Answer (pick one from the list):
4) To win I don't know if you need to kill or reach/touch all orange
objects

--

Do you lose if orange objects die?
1) You lose if all orange objects die or disappear
2) You don't lose if all orange objects die or disappear
3) I don't know if you would lose if all orange objects die or
disappear

Reasoning: ......................

Answer (pick one from the list):
2) You don't lose if all orange objects die or disappear

### Example Game 2

Message from the player:
"You can shoot brown objects. Brown objects kill gold objects when
they touch them. Gold objects transform purple objects into lightred
objects when they touch them. You transform lightred objects into
purple objects when you touch them. You lose if all purple objects
are transformed into lightred objects. You win if you survive long
enough."

--

What happens to gold objects when they collide with brown objects?
1) nothing happens to gold objects when they collide with brown
objects
2) gold objects die when they collide with brown objects
3) gold objects get transformed when they collide with brown objects
4) gold objects steal resource from brown objects when both collide
```

```
5) gold objects die if they don't have enough resources when they
collide with brown objects
6) I don't know / something else

Reasoning: ......................

Answer (pick one from the list):
2) gold objects die when they collide with brown objects

--

What happens to purple objects when they collide with gold objects?
1) nothing happens to purple objects when they collide with gold
objects
2) purple objects die when they collide with gold objects
3) purple objects get transformed when they collide with gold objects
4) purple objects steal resource from gold objects when both collide
5) purple objects die if they don't have enough resources when they
collide with gold objects
6) I don't know / something else

Reasoning: ......................

Answer (pick one from the list):
3) purple objects get transformed when they collide with gold objects

--

What happens to lightred objects when they collide with gold objects?
1) nothing happens to lightred objects when they collide with gold
objects
2) lightred objects die when they collide with gold objects
3) lightred objects get transformed when they collide with gold
objects
4) lightred objects steal resource from gold objects when both
collide
5) lightred objects die if they don't have enough resources when they
 collide with gold objects
6) I don't know / something else

Reasoning: ......................

Answer (pick one from the list):
6) I don't know / something else

--

How do lightred objects behave?
1) lightred objects cannot move
2) lightred objects do not move and disappear after a certain time
3) lightred objects regularly spawn/generate other objects
4) lightred objects can be pushed
5) lightred objects move along one axis
6) lightred objects move along one axis and regularly spawn/generate
objects
7) lightred objects chase or flee another object
8) lightred objects move randomly
9) lightred objects are portals
10) I don't know

Reasoning: ......................

Answer (pick one from the list):
10) I don't know
```

```
--

Do you need to kill brown objects to win?
1) To win you need to reach/touch brown objects
2) To win you need to kill all brown objects
3) To win you don't need to kill all brown objects
4) To win I don't know if you need to kill or reach/touch all brown
objects

Reasoning: ......................

Answer (pick one from the list):
3) To win you don't need to kill all brown objects

--

Do you lose if purple objects die?
1) You lose if all purple objects die or disappear
2) You don't lose if all purple objects die or disappear
3) I don't know if you would lose if all purple objects die or
disappear

Reasoning: ......................

Answer (pick one from the list):
1) You lose if all purple objects die or disappear

--

Do you lose if lightred objects die?
1) You lose if all lightred objects die or disappear
2) You don't lose if all lightred objects die or disappear
3) I don't know if you would lose if all lightred objects die or
disappear

Reasoning: ......................

Answer (pick one from the list):
2) You don't lose if all lightred objects die or disappear

### Task

Message from the player:
{MESSAGE}

[QUESTION]
[CHOICES]

Reasoning: ......................

Answer (pick one from the list):
[CANDIDATE ANSWER] # we compute the likelihood of all possible
candidate answer tokens
```

**Prompt for the pure-LLM baseline**

```
{basicstyle=\ttfamily\small}
You are trying to solve a video game you have never player before.

### Information about the game:
```

– The game contains colored square objects of size 1
– It is defined by:
  – one object type for each object color (eg a missile, a random NPC,
   an immovable object, a pushable object, a portal, etc.)
  – interactions for each possible pair of object color (eg no
  interaction, one dies, one is transformed into the other, one pushes
   the other, one steps back, etc.). These define what happens when
  objects of these two colors collide.
  – wining conditions: you may win by surviving long enough or when
  you kill all objects from a specific list of colors (eg kill all
  green and white to win)
  – losing conditions: you may lose if you don't win fast enough, or
  if all objects of a specific color die (eg you lose if either all
  red die or all blue die)
– Each object color corresponds to a different type of object, but
all objects of one color have the same type and behave the same way
– Each pair of object colors can have at most 1 type of interaction
– Some objects (color1) can transform certain objects (color2) into
yet other objects (color3).
– Possible ways to act in the game are to touch objects, push objects
 or shoot. Yourself, the objects you push, or shoot can interact with
 other objects. Sometimes they will push them, kill them, transform
them. Sometimes nothing happens.

### State description:
– You will be shown your recent actions and the corresponding history
 of game states, as well as object movements between these states
– A game state is composed of lists of object positions grouped by
object color
– Positions are indicated as (x, y) where x is between x=0 (LEFT) and
 x=max_x (RIGHT), and y is between y=0 (TOP) and y=max_y (BOTTOM).
– When x increases, objects move to the right, when x decreases,
objects move to the left. When y increases, objects move downwards,
when y decreases, objects move upwards. Moving UP decreases y,
movement = (0, -1).
– Objects have size 1. They collide when delta_x or delta_y is below
1.
– Look for wall positions, you cannot move through walls and shouldn'
t try to move in the direction of a wall.

### Analysis:
– Observe the recent history of states and actions in the environment
– Observe your scratchpad where you store your analysis of the game
and strategy to solve it
– If your previous action was unsuccessful, try to understand why.
– If you won or lost in the previous episode, try to understand why.
– Be careful about picking the right action to go where you want to
go.
– Try to predict the movements of important moving objects so you can
 either catch or avoid them with your next action.
– Analyze your scratchpad, should you rephrase it to be more compact
and up to date?
– Analyze both to update your understanding of the game, your
strategy, and prepare for action.
– Don't forget to plan for your next action. You might want to
explore the world and touch, push or shoot objects, or you might want
 to plan towards the goal.

### Scratchpad update:
– Decide whether to append information to the scratchpad or to
replace the text in the scratchpad by new text.
– The scratchpad should contain everything you know about the
environment: possible interactions, what kills you, what you think
the goal of the game is, what makes you lose, etc.

- The scratchpad should contain your current plan to explore or solve
 the game.
- The scratchpad is the only memory that will be transmitted to the
next decision making step, write down everything you need.
- To append text, use <append> text to append goes here </append>
- To replace text, use <replace> new text goes here </replace>
- When replacing the scratchpad, all replaced text will be forgotten,
 be careful.
- No need to write down observations in the scratchpad, you will be
provided with the last observations every time you are asked to
generate a new action.
- Write your goal in the scratchpad so you can keep it in mind at the
 next step.
- The scratchpad has a maximum size of 400 words, you should
sometimes use the <replace> command to summarize its content and keep
 it concise.
- You start each new episode with the scratchpad from the last
episode. It will contain useful information about the game you're
playing (eg how you can die, win, or important game dynamics), but
may also contain outdated information (eg object positions, or
previous strategy from the last episode).
- When reaching the end of the episode, rewrite the scratchpad to
transmit information about what you learn so you can do better in the
 next episode.

### Actions:
- Possible actions are: UP, DOWN, LEFT, RIGHT, SPACE_BAR and NOOP.
- UP decreases y (0, -1), DOWN increases y (0, +1), LEFT decreases x
(-1, 0), RIGHT increases x (+1, 0), SPACE_BAR and NOOP do not move
you.
- SPACE_BAR let's you shoot / spawn objects in some games (not all).
- Try SPACE_BAR sometimes to see if anything happens, and write to
the scratchpad what happens.
- Make sure your actions get you closer to your goal. It's easy to
pick the wrong action and get further away, pay attention!
- Every time you pick an action that is not NOOP, you will have to
wait 4 environment steps before picking a new action. If you select
NOOP you can select another action at the next environment step.

### Answer format:
- Analyze what you know about the game and the history of recent game
 states
- Update the scratchpad
- Select the action to execute next

Your response should be formatted in this way:

% first the analysis
<analysis>
write here your reflection and analysis of available information,
what you understand about the game, your plan to update the
scratchpad and your strategy for picking the next action.
</analysis>

% then update the scratchpad with either
<append>
text to append to the scratchpad goes here
</append>
% or
<replace>
text to replace the scratchpad with goes here
</replace>

% finally pick an action among UP, DOWN, LEFT, RIGHT, SPACE_BAR, NOOP

```
<act>
action goes here
</act>

### Example

#### Recent history
% game history appears here

#### Old scratchpad
Orange objects can kill you so you should watch out. Shooting red
objects seem to kill them. I'm not sure what the goal is yet. I can
push blue objects but not purple ones.
Strategy: I should try to touch the yellow object to see what happens,
 maybe I need to reach it?

#### Example answer

<analysis>
It seems orange just killed the blue object? I should add this to the
 scratchpad. I'm still relatively far from the yellow object I should
 keep moving in that direction.
My next move should be UP.
</analysis>

<append>
Orange kill blue when they touch.
</append>

<act>
UP
</act>
```

## K  BASELINE COMPARISONS

Figure 10 shows the performance of the deep RL baseline when run for up to 2,000 episodes.

For completeness, we considered whether similar long-horizon runs should be performed for the LLM agent. This is not feasible for several reasons. First, the LLM agent is extremely expensive to execute: it produces a chain-of-thought plan at every time step, and concatenating these traces quickly saturates the context window, preventing the model from carrying information across many episodes. Second, longer training is unlikely to change outcomes. Inspection of the model's reasoning traces shows persistent difficulties in (1) inferring rules from observations, (2) forming coherent multi-step plans, and (3) executing those plans reliably. These limitations mirror recent findings on LLM performance in long-horizon video-game benchmarks (e.g., Balrog AI (Paglieri et al., 2024)), where LLMs perform poorly even when given full rule descriptions. VGDL games are even more challenging, as each new game introduces novel rules that must be inferred from scratch. For these reasons, long-horizon rollouts are not expected to materially improve the LLM baseline.

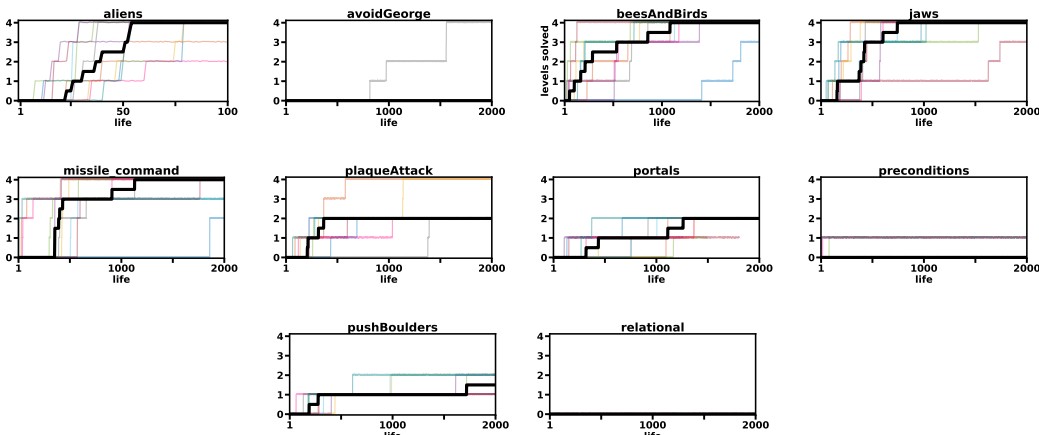

Figure 10: Deep RL performance with extended training horizons (2,000 episodes per game, double DQN). Aliens is solved within the first 50 lives, as it brings a dense straightforward signal (+1 for each alien shot). *beesAndBirds*, *jaws* and *missileCommand* can also be solved within the first 2,000 episodes, as they also demonstrate relatively straightforward reward signals. In all other games, DDQN fails to explore sufficiently, e.g., solving none of the levels in *relational*, *preconditions*, and *avoidGeorge* due to sparse rewards. These results illustrate the difficulty of learning to solve unknown VGDL games without explicit world modeling.

