# OpenReview forum: "Language and Experience: A Computational Model of Social Learning in Complex Tasks"
_ICLR.cc/2026/Conference — ICLR 2026 Poster_

### Official Review · Reviewer_YLjG · 2025-10-26

**Soundness:** 3
**Presentation:** 3
**Contribution:** 3
**Rating:** 6
**Confidence:** 3

**Summary:**

The authors propose a Bayesian model by which agents use both linguistic guidance and direct experience to develop a world model of their environment. Their proposed agent updates their distribution of world models based on compatibility with game observations and linguistic guidance from a model, weighted by probability assigned by an LLM-based speaker model. Experiments on the VGDL suite of games shows that their agents learn significantly more quickly than Deep RL and LLM agent-based baselines, and that linguistic guidance greatly accelerates learning when added to direct experience. They also find that model-written guidance can also improve learning for human participants, although both models and humans learn better from their own guidance.

**Strengths:**

- Thorough theoretical framing of the important problem of learning jointly from language and experience, from which the proposed model clearly follows.
- Results across a variety of game settings show that the model learns much more efficiently than Deep RL and LLM agent-based approaches, suggesting that the model does, in fact, approximate human learning mechanisms better than existing methods.
- Rigorous study of the impact of human vs model guidance, as well as differences in communication styles, motivates some pretty interesting future directions related to collaboration in human+AI communities.

**Weaknesses:**

- The baselines used are both highly limited within the time horizon given, which limits interpretation of results. It would be useful to run the pure LLM and DQN baselines over a longer horizon until some learning happens, to get a better sense of how much more efficient the proposed model is. This could also help us better understand the relative gains from incorporating linguistic guidance in the model vs pure LLM settings, which would strengthen the argument that the new agent better models human learning improvements from linguistic guidance.
- As the authors note, they evaluate agents over a very limited set of game environments. Fully open-ended environments would be out of scope of the paper, but it’s not clear to me that the games used are significantly more complex than the environments used in past work (e.g. the 2024 Zhi-Xuan work cited) that are dismissed as overly simplistic.

**Questions:**

- It’s not clear to me from the citation given in 3.4 why language-conditioned RL methods require thousands of episodes to learn language-policy mappings, especially since the citation is from 2019 and more recent methods use LLMs to become more sample efficient (e.g. "Language instructed reinforcement learning for human-ai coordination”, Hengyuan Hu et al, ICML 2023). Since this seems to be more relevant to the proposed method than the Deep RL or LLM Agent baselines, it would be useful to spend more time justifying the decision to not benchmark against them.
- I’d recommend including median hourly pay for the crowdworkers, either in Appendix G or Section 4.

---

> ### Author Response · Authors · 2025-11-20
> **First answer to Reviewer YLjG**
>
> We thank the reviewer for the thoughtful and encouraging feedback. We appreciate the recognition of the theoretical framing, the strong empirical results across tasks, and the relevance of our findings for future human–AI collaboration research. We address each remaining concern below.
>
> **1. Baselines and learning horizon**
>
> We agree that DQN learns slowly, which is part of our argument. Our model aims at capturing human-like sample efficiency,  and how language shapes both performance and exploration patterns. A central finding (consistent with prior work such as Tsividis et al., 2021) is that achieving human performance and efficiency requires structured, executable world models and model-based planning.
>
> We did run DDQN for much longer training horizons (2,000 episodes per game). As shown in the new Appendix Section K, extended training lets DQN solve a few games with dense or straightforward reward signals (e.g., aliens, beesAndBirds, jaws), but completely fails to solve games with sparse or relational structure (relational, preconditions, avoidGeorge), even after thousands of episodes. These results reinforce our main claim: without explicit world-model structure, exploration remains inefficient, and reward-driven trial-and-error alone is insufficient to capture human-like learning performance and sample efficiency in these environments.
>
> We also agree that, in principle, longer horizons could be explored with LLM agents. However, the LLM agent is extremely expensive to run: it generates a chain-of-thought plan at every single step. Appending these contexts quickly fills the context window, limiting the capacity of the agent to learn across many episodes. More importantly, longer horizons are not expected to qualitatively change results, as we observed from reasoning traces their inability to 1) infer rules from observations, 2) form coherent plans, 3) carry on these plans. These observations are consistent with LLM performance on long-horizon games (e.g. Balrog AI), where LLMs achieve poor performance. VGDL is even harder, as each new game involves new rules that the LLM must explore and infer first, while Balrog games are fully described to the agent and most probably part of LLM training corpora. Thus, longer rollouts would not be expected to help the LLM agent perform significantly better.
>
> We added a new section in the Appendix showing results of longer deep RL runs and discussing running LLM baselines for longer.
>
> **2. Complexity of our environments vs. prior work (e.g., Zhi-Xuan et al. 2024)**
>
> We appreciate the chance to clarify this. The environments in Zhi-Xuan et al. (2024) and related Bayesian ToM work differ fundamentally from ours:
>
> * They require goal inference, but the transition function is known and is fully deterministic. There is no other agent to model, the goal space is limited
> * Planning can be done with PDDL-style symbolic planners
> * The hypothesis space does not include causal world-model uncertainty, simply uncertainty over possible goals
> * The evidence is not collected by the agent’s own experience but is observed from pre-collected, hand-defined trajectories.
>
> In contrast, VGDL tasks require:
> * inferring object types,
> * inferring possible pairwise interaction rules,
> * inferring win/loss conditions,
> * performing model-based planning in environments with multiple hazards, transformations, and stochastic elements.
> * letting agents collect their own data as they learn about the world
>
> From a cognitive-science perspective, the Zhi-Xuan environments isolate goal inference; our environments require full structural causal inference plus planning. We now clarify this distinction explicitly at the end of Appendix A.
>
> **3. Benchmark against modern language-conditioned RL**
>
> We thank the reviewer for their suggestion. The method of Hu et al. maps language to tabular Q-values and uses this as a prior over discrete policies. This works in small state spaces with short horizons, but it does not transfer to VGDL-scale tasks where policies require large neural function approximators, long-range planning, and structured representations of object dynamics. In these domains, the gap between linguistic abstractions and the low-level policy representation becomes too wide for a direct language→policy mapping to be effective. Our approach instead uses language to constrain world-model inference, not policy learning, which makes it applicable in settings where policies are non-tabular, high-dimensional, and require symbolic structure to interpret linguistic guidance.
>
> **4. Hourly pay for crowdworkers**
>
> Thank you for the suggestion. We have added the median hourly pay rate to Appendix G.

---

> > ### Comment · Reviewer_YLjG · 2025-11-26
> >
> > Thanks for your thorough response. This (as well as the additional crowdworker information mentioned elsewhere) addresses my concerns, and I'm changing my score from 6 to 8 accordingly.

---

### Official Review · Reviewer_LmtC · 2025-10-29

**Soundness:** 3
**Presentation:** 3
**Contribution:** 3
**Rating:** 6
**Confidence:** 2

**Summary:**

This paper develops a method to integrate natural language advice into planning as well as transform experience into natural language advice. The authors demonstrate that both human-generated and model-generated advice improve sample efficiency in learning a suite of tasks.

Overall, I recommend acceptance because the method facilitates effective knowledge transfer between humans and models, providing a valuable avenue for future work on human-AI co-learning. However, because the method is restricted to the VGDL game, it could be difficult to scale to real-world tasks.

**Strengths:**

The results convincingly show that advice, whether human-generated or model-generated, improves learning performance. The evaluation is conducted across multiple tasks, demonstrating the robustness of the approach. The method is well-motivated and the results are intuitive (advice improves learning, humans can learn better from human advice than model advice). The generational learning experiments provide additional evidence that the model-generated advice is effective, strengthening the paper's claims.

**Weaknesses:**

As the authors acknowledge, the method is currently limited to the VGDL game space. Scalability to more complex, diverse, or realistic environments could be difficult.

The prompts for integrating natural language advice could potentially have information leak (see questions below), which may bias the results.

**Questions:**

**Questions for clarification:**

You mention "prompting the LM with the received message L and instructing it to answer multiple-choice questions about specific VGDL rules." Wouldn't this give the model an advantage over the human or pure LM baseline because the multiple choice answers already narrow what the feasible rules are? Is it possible to evaluate the method without this step by sampling candidate rules directly from the LM?

---

**Minor comments:**

- It would be helpful to add effect sizes to the reported results.
- The authors state that "We compare our approach to three baselines: 1) Oracle: a model that plans to solve the game using ground-truth game rules, (Line 467) " but the oracle baseline is not shown in the main results.
- In the paragraph "Variability across games," the authors conclude that "These patterns suggest that social learning strategies like selecting high-performing teachers" (line 400), but I don't think this conclusion follows from the results. The results don't imply that higher-performing participants gave better advice.
- The authors state that "We show more examples of human- and model-generated messages in Appendix Section I and on our website (Line 385)." The website only has some game demos. I couldn't find the examples of human and model-generated messages.

---

**Suggestions for improvement (not factored into score):**

The authors state "Human-generated messages often include game abstractions, high-level strategies, and planning heuristics that our model currently cannot leverage (Line 467)." While the authors do provide examples of human-generated and model-generated messages in the appendix, it would be helpful to more systematically categorize the differences between human and model-generated messages. This would be beneficial for further work on human-AI collaboration.

---

> ### Author Response · Authors · 2025-11-20
> **First answer to Reviewer LmtC**
>
> We thank the reviewer for the positive and encouraging assessment. We appreciate the recognition of the clarity of our motivation, the robustness of the results across tasks, and the potential of our framework for future research on human–AI co-learning. We address each remaining concern below.
>
> **1. Scalability beyond VGDL**
>
> Our goal is not to solve arbitrary real-world inference but to model how humans jointly integrate linguistic and experiential evidence (introduction, §3). VGDL provides a cognitively grounded space of executable causal theories, consistent with prior work on human causal learning and structured theory induction (Tsividis et al., 2021; Tessler et al., 2021). We have added text emphasizing this framing (Discussion §1) .
>
> Scaling to more open-ended environments is an important future direction for AI, and we expanded the discussion to note how recent advances in program synthesis and LLM-based code modeling could support this (§6). The fact that humans routinely build rich executable simulations of complex processes (physics engines, game environments, scientific models), suggests that this is both a valuable and a feasible future direction.
>
> **2. Potential information leak in LM-guided multiple-choice prompts**
>
> The reviewer raises an important point. The purpose of the multiple-choice questions is not to provide additional rule information unavailable to other agents—it is to map natural language to a predefined, shared hypothesis space over VGDL rules. All agents work within this same structured space:
>
> * Humans are introduced to the types of possible mechanics through the tutorial level (movement, collision effects, transformations, shooting, etc.).
> * Deep RL agents do not operate over symbolic world models, so multiple-choice rule interpretation does not advantage or disadvantage them—they cannot use rules at all.
>
> The pure LLM baseline is explicitly informed of the same possible VGDL primitives through its prompt (Appendix J), which ensures a fair comparison.
>
> The key point is that the multiple-choice format does not narrow the space beyond what is already fixed by the hypothesis class; it simply provides a reliable way to query how well a given message supports candidate rule edits during inference. The multiple-choice interface ensures consistency with the structured hypothesis space. It does not bring an advantage over other baselines.
>
> **3. Effect sizes**
>
> We already report effect sizes as Δ(nAUCs) (e.g., in Section 5.2), which quantify the size of the improvement in a normalized performance metric (area under the curve, see definition in Section 4). We added a brief clarification to ensure this is clear to readers.
>
> **4. Oracle baseline not shown**
>
> The oracle baseline appears in the game-specific individual learning plots in Appendix Section B. It was indeed missing from Figure 3, which is now corrected. It represents a planner with access to the ground-truth world model. Because the oracle is not affected by receiving advice (its beliefs never change), it appears only in the individual learning plots and not in the social learning figures.
>
> **5. Conclusion about teacher selection (line 400)**
>
> We agree that our original wording could be misinterpreted. We revised the sentence to avoid implying that higher-performing participants necessarily produced higher-quality advice. The intended meaning is that a message generated by a certain teacher might be more informative or pedagogical than others, thus leading to higher-performing learners, and learners might want to select teachers that tend to generate these.
>
> **6. Missing message and systematic comparison of human vs model messages**
>
> Thank you for pointing this out. We have added the human- and model-generated messages referenced in the text to the project website.
>
> We agree comparing human and model-generated messages is an interesting direction. Our paper already performs a structured analysis of human messages (manual coding of dynamics, win/loss conditions, and errors). This said, model-generated messages tend to be highly structured, literal, and free of analogies or metacognitive reflections; features that emerge from the structure of the advice generation prompt (cf Appendix Section J). As a result, it is for now more interesting to look at human messages (more varied) than machine messages (more formatted).

---

> > ### Comment · Reviewer_LmtC · 2025-11-23
> >
> > I appreciate the authors' detailed response, including discussion on scalability and clarification on the role of MC rule interpretation. With the added clarification, I am convinced that the baselines are fair given that humans, RL agents, and LLMs operate at different levels mechanistically. My minor comments have also been addressed. Therefore, I increase my score further and recommend acceptance.

---

### Official Review · Reviewer_46yR · 2025-10-31

**Soundness:** 3
**Presentation:** 3
**Contribution:** 3
**Rating:** 6
**Confidence:** 3

**Summary:**

This paper explores how language and experience jointly support social learning in humans and computational agents. The work takes the view of human social learning as probabilistic inference over world models. Empirically, they study human participants and Q-learning agents performing ten games, examining how linguistic guidance shapes exploration and accelerates learning. A pretrained large language model (LLM) is used both to generate advice for other agents and to interpret linguistic input as evidence.
The results show that linguistic advice improves learning efficiency for both humans and models. Human participants learn equally well from model- and human-generated advice, whereas models benefit more from model-generated advice. The authors attribute this to differences in communication styles — human messages often include metacognitive reflections and analogies that LLMs find difficult to parse.
The paper also investigates iterated (generational) learning, where agents interact with the environment for limited “lifetimes” and pass advice to subsequent agents. Performance generally increases across generations, though with some regressions depending on the game.

**Strengths:**

* The writing is clear and engaging.
* The combination of behavioral experiments with humans and artificial agents is very creative.
* The results on cross-generational knowledge transfer are intriguing and connect well to theories of cumulative culture and iterated learning.
* The work highlights a bidirectional exchange between humans and models — both learn from each other, which is a timely topic to consider.

**Weaknesses:**

* _Framing_ The title and framing promise a computational model of social learning, but the core of the paper is some empirical observations of social learning among and between agents and humans. The authors should consider revising the title or clarifying that the contribution is primarily empirical and conceptual, not a new formal model.
* It is unclear what are the structured world models under the claimed “joint probabilistic inference over structured world models.”
* The claim that the LLM performs Bayesian inference is not supported by direct evidence. The LLM generates or interprets advice, but it is not shown to approximate a Bayesian posterior or perform probabilistic updates.
* The results for iterated (generational) learning vary substantially across games (e.g., improvement for Bess and Birds, but not Plaque Attack or Missile Command). It remains unclear why certain tasks show cumulative improvement while others regress, and what properties of the games modulate this effect.
* Figure 6 (the Aliens game) may not be a good test of generational learning, as performance saturates quickly, leaving no room for measurable improvement.

**Questions:**

1. What exactly constitutes the “computational model” here — is it the Q-learning setup augmented with linguistic input, or the LLM itself viewed as a Bayesian speaker–listener model?
2. Is there any quantitative or behavioral evidence that the LLM performs probabilistic inference rather than pattern-based generation?
3. Why do some forms of advice promote generational improvement while others do not? Are there identifiable linguistic or structural features  that predict transfer success?
4. Given that human messages contain analogies and metacognitive strategies that LLMs fail to parse, could fine-tuning on human teaching data improve model interpretability and alignment?

**Details Of Ethics Concerns:**

A substantial part of this study is about human behavior. I was looking through the SI and did not find a paragraph about participant recruitment and compensation. And this is quite important to report in the paper of the main contribution comes from observations from behavioral studies.

---

> ### Author Response · Authors · 2025-11-20
> **First answer to Reviewer 46yR**
>
> We thank the reviewer for the constructive and thoughtful assessment. We appreciate the recognition of the clarity of the writing, the integration of human and model experiments, and the interest in our generational-learning results. We address each point below.
>
> **1. Framing: “computational model” vs. “empirical observations.”**
>
> The paper’s main contribution is a computational model of how agents integrate linguistic and experiential evidence. It is not an empirical study nor “Q-learning plus language”: our model does not contain Q-learning. It performs Bayesian inference over executable world models, using (i) a structured hypothesis space of VGDL programs, (ii) Bayesian updates combining experience likelihood P(E|T), language likelihood P(L|T), and a simplicity prior, (iii) particle filtering with MCMC rejuvenation, and (iv) planning over the MAP theory. Human data serve to evaluate this model. These different parts are all detailed in Section 3.
> This also addresses the reviewer’s questions about what constitutes the computational model and whether the LLM performs Bayesian inference (it does not). The LLM only serves as a proposal distribution and an observation distribution (speaker model). The overall inference architecture implement Bayesian inference, not the LLM itself.
>
> **2. “What are the structured world models?”**
>
> Structured hypotheses are executable VGDL programs specifying object types, pairwise interactions, rewards, and win/loss rules. Combined with the VGDL compiler, they define full causal transition models, reward and termination functions so a full MDP that the player can play in its head; see descriptions in Section 2 and the paragraph a space of possible worlds in Section 3. These representations are “structured” in the cognitive-science sense (program-like generative theories), not neural dynamics models. We clarified this in Section 3: “World models are programs specifying the transition function and reward structure, while inference consists of updating a posterior distribution over these executable programs.”
>
> **3. “The claim that the LLM performs Bayesian inference is not supported.”**
>
> We agree. The paper does not claim this. The agent performs approximate Bayesian inference; the LLM only provides (i) proposal distributions for rule edits and (ii) an approximate speaker model for P(L|T). The LLM does not maintain beliefs or perform posterior updates.
>
> **4. Variation in iterated-learning results across games.**
>
> As the reviewer points out, and is discussed in 5.4, generational improvement is observable only when Generation-1 performance is not at ceiling. In such games, improvement is consistently observed. Two games show greater variability:
> • relational: many interdependent transformation rules; with only two lives, a single wrong push can irreversibly block progress, producing volatility.
> • plaqueAttack: two distinct survival strategies (eliminate attackers vs. reconquer bases). Messages often describe only one, which can mis-steer later generations.
> Games with simpler, linguistically compressible mechanics (e.g., avoidGeorge, beesAndBirds) transmit more robustly. We added a new paragraph in 5.4 to discuss these points.
>
> **5. Why some advice produces generational improvement.**
>
> Sections 5.2 and 5.3 already address part of this: we hand-coded human messages and found that longer messages and correct win-condition information strongly predict performance. Tasks whose key dynamics can be succinctly verbalized benefit more from guidance.
>
> We agree that a deeper linguistic analysis—e.g., characterizing which specific abstractions or structural features in messages support robust generational transfer—would be valuable. This is an active research direction, and prior work using human iterated-learning chains in VGDL environments (Tessler et al., 2021) provides relevant methodology for such fine-grained message analyses. We have added a brief comment in the Discussion to highlight this as a promising extension (5th paragraph).
>
> **6. Could fine-tuning help models parse human messages?**
>
> Possibly for interpreting metaphors or analogies, but fine-tuning would not allow the model to exploit metacognitive or strategic advice: the planning module, world-model representation, and goal structure are fixed and not modifiable by language. We comment in the Discussion that future work could map language to structured symbolic goals (e.g., PDDL-like forms), but this is beyond the present scope.

---

> ### Author Response · Authors · 2025-11-26
> **Answer to the ethics flag**
>
> Our previous answer did miss the comment in the ethics part of the review. We thank the reviewer for raising this point and asking for additional details.
>
> Some details about participant recruitments (number of participants, selection criterion, recruitment platform) were already present in the first version of the paper (Section 4, paragraph *Participants*). This said, the original version did omit a number of details, that we now added to Appendix G now renamed *Human data collection: instructions and participant recruitments.* Participants were all adult English-speakers and the median time spent and median hourly pay (49min, $10/h).

---

### Official Review · Reviewer_kQQx · 2025-11-01

**Soundness:** 3
**Presentation:** 3
**Contribution:** 3
**Rating:** 8
**Confidence:** 4

**Summary:**

The paper presents a framework that models human social learning as joint probabilistic inference over structured world models, integrating both linguistic and sensorimotor information. The core method used is to transform a pretrained language model into a probabilistic model of human advice, enabling agents to generate and interpret guidance. Experiments across ten video games show that linguistic input enhances exploration efficiency, safety, and learning speed. Iterated learning experiments further demonstrate knowledge accumulation and transfer between humans and models, supporting more effective human–AI collaboration.

**Strengths:**

(1) The paper is well-written, with clear motivation and an accessible presentation. The formulation of joint inference over experiential and linguistic guidance through Bayesian updating is both insightful and valuable.

(2) The human evaluation experiments on building better learning partners are interesting and effectively demonstrate the role of linguistic guidance in improving video game performance. Moreover, the study’s extension to bidirectional communication between humans and models provides an elegant and symmetric perspective on social learning.

**Weaknesses:**

(1) Limited generalization: The current modeling approach is evaluated across different video games. However, in real-world agentic environments, the belief space can be far more complex, posing additional challenges for both modeling and linguistic guidance. As a result, the proposed method may have limited applicability beyond specific domains.

(2) Model selection discussion: It would be valuable to include an analysis of different LLM choices. Stronger models may offer more effective guidance for both LLMs and human experts—or potentially the opposite. An additional study could examine whether the same phenomenon shown in Figure 5 persists when the guidance-providing model and the student model belong to different model families.

**Questions:**

The difficulty of different games can vary significantly depending on the complexity of their underlying theories and gameplay mechanics. It would be helpful to clarify whether the size of the candidate theory set varies across games and how much this variation affects performance. I am also interested in more details about the selection of M (candidate theory), particularly the reasoning behind choosing 20.

---

> ### Author Response · Authors · 2025-11-20
> **First answer to reviewer kQQx**
>
> We thank the reviewer for the positive and thoughtful evaluation. We appreciate the clear recognition of the paper’s strengths, particularly the value of the Bayesian formulation for integrating language and experience and the interest in the bidirectional human–model experiments. We address each remaining concern below.
>
> **1. Limited generalization**
>
> Our goal is not to solve arbitrary real-world inference but to model how humans jointly integrate linguistic and experiential evidence (introduction, §3). VGDL provides a cognitively grounded space of executable causal theories, consistent with prior work on human causal learning and structured theory induction (Tsividis et al., 2021; Tessler et al., 2021). We have added text emphasizing this framing (Discussion §1) .
>
> Scaling to more open-ended environments is an important future direction for AI, and we expanded the discussion to note how recent advances in program synthesis and LLM-based code modeling could support this (§6). The fact that humans routinely build rich executable simulations of complex processes (physics engines, game environments, scientific models), suggests that this is both a valuable and a feasible future direction.
>
> **2. Model selection and LLM choices**
>
> We appreciate the reviewer’s suggestion to analyze the impact of different LLM choices. This is indeed an interesting question. Our current experiments use a single LLM for the speaker and the speaker model and we agree that using different models could lead to differences in pedagogical style or informativeness. At the same time, we already demonstrate a stronger form of generalization: the model successfully interprets guidance produced by humans, whose language is far more variable than that of any alternative LLM family. This shows that the speaker model used for inference is robust to large cross-domain mismatches in communication style.
>
> While an additional sweep across LLM families is beyond the scope of the present study, we have added a sentence in the Discussion noting this as a promising extension, particularly for exploring how model-specific communication styles influence social learning.
>
> **3. Variation in candidate theory space across games**
>
> We thank the reviewer for this question. The theoretical size of the hypothesis space does differ across games because it is a function of the number of object types in the game (each additional object increases the combinatorial possibilities for object types, interactions, and win/loss conditions). This scales up very quickly with the number of objects, reaching more than 10^30 possible theories with the games containing the smallest number of objects (5).
>
> However, the effective difficulty of inference—and ultimately the model’s performance—correlates less with the raw size of the hypothesis space and more with how easy it is to obtain disambiguating evidence. Games that provide clear, frequent feedback signals (e.g., object collisions with deterministic outcomes) are learned quickly even if they have large hypothesis spaces. Conversely, games such as “Relational” can be harder despite having fewer objects, because the critical structure involves discovering sparse, compositional object-interaction rules. We added an explanation of these points in the paragraph Size of the search space and game difficulty of Appendix C.
>
> **4. Why M = 20 candidate theories?**
>
> The number of particles involves a trade-off:
>
> * Exploration: too few theories lead to under-representation of uncertainty, reducing the model’s ability to identify informative subgoals and to explore sufficient alternative variations in the space of possible theories.
> * Computation: each Metropolis update requires updating all particles and computing language likelihoods via the LLM, which is the computational bottleneck of the approach.
>
> Empirically, 10 particles was faster but sometimes failed to discover high-posterior theories quickly enough for the agent to survive, while 20 ensured a more robust and fast exploration of the theory space. More particles might slightly improve the performance of the inference, but will make the system slower to run and experiment with. We now justify this choice in Appendix D.

---

### Official Review · Reviewer_GnG7 · 2025-11-05

**Soundness:** 3
**Presentation:** 3
**Contribution:** 3
**Rating:** 6
**Confidence:** 4

**Summary:**

This paper proposes a novel computational framework for social learning in AI, which rigorously integrates linguistic guidance and direct sensorimotor experience through joint probabilistic inference. The core innovation is leveraging a pre-trained Large Language Model (LLM) as a probabilistic model of human advice ($P(\text{Language} | \text{Model})$). By treating language as evidence to infer underlying structured, executable world models, the agent achieves superior sample efficiency and rapid, safe knowledge acquisition in complex tasks, significantly outperforming conventional learning methods.

**Strengths:**

1. Novelty in Theoretical Framework: The paper introduces an innovative computational model that treats language not merely as an input feature, but as probabilistic evidence for Bayesian inference over potential world models. This is a theoretically rigorous approach to language grounding.

2. High Sample Efficiency: By leveraging linguistic guidance as a strong prior, the method significantly reduces the hypothesis space, leading to substantially faster learning and higher sample efficiency compared to purely experiential methods.

3. Emphasis on Structured Knowledge: The model focuses on inferring structured, executable world models (rules and mechanisms), which inherently promotes generalization across tasks and provides better interpretability than learning simple policies or value functions.

4. Robust Integration: The probabilistic nature of the framework allows for the principled integration of two heterogeneous data sources (noisy language and imperfect experience), making the system robust to ambiguity and uncertainty in human advice.

5. Applicability to Theory of Mind: By converting the LLM into $P(\text{Language} | \text{Model})$, the agent performs a form of reverse inference, effectively modeling the human advisor's underlying knowledge or "mind," which is a powerful concept in social learning.

**Weaknesses:**

1. Scalability Challenge in Model Space: While the framework operates over "structured world models," the computational feasibility of performing inference (or even representation) over the vast space of possible models ($\text{Model}$) in real-world, high-dimensional, or continuous state environments remains a major concern. The paper needs to better address how efficient search or approximation is maintained.
2. Experimental Complexity Limitation: Although the title claims to address "Complex Tasks," the typical experimental setups (often discrete environments like Block-World or Grid-World, based on the context of similar research) may not fully demonstrate the method's effectiveness and scalability in truly unstructured, high-fidelity, or continuous environments, limiting the generalizability claim.
3. Insufficient Comparison to Language Baselines: The comparison with existing methods may be incomplete. Specifically, the paper often omits a direct comparison with state-of-the-art deep reinforcement learning (DRL) techniques that use language as an auxiliary input feature or recent LLM-based planning methods (e.g., Chain-of-Thought planning), which could be a strong alternative baseline for incorporating guidance.
4. Technical Validity of the LLM Probability Approximation: The paper's core mechanism, the Language Likelihood $P(\text{Language} | \text{Model})$, is approximated using the uncalibrated generation probability of a large proprietary model (LLAMA-3.1-70B) conditioned on a constructed prompt: $P(L|T) \approx P_{LM}(L|\text{prompt}(T))$. This approach raises concerns about its technical validity. LLM generation probabilities are generally not guaranteed to be accurately calibrated for strict Bayesian inference, which is central to the framework. The reliance on prompt engineering for a fundamental probabilistic term introduces a fragility that needs more rigorous theoretical justification or empirical validation of the approximation accuracy.

**Questions:**

See above.

---

> ### Author Response · Authors · 2025-11-20
> **First answer to Reviewer GnG7**
>
> We thank the reviewer for the thoughtful and constructive assessment. We appreciate the clear articulation of the paper’s strengths, and we address remaining concerns below.
>
> **1. Scalability Challenge in Model Space**
>
> Our goal is not to solve arbitrary real-world inference but to model how humans jointly integrate linguistic and experiential evidence (introduction, §3). VGDL provides a cognitively grounded space of executable causal theories, consistent with prior work on human causal learning and structured theory induction (Tsividis et al., 2021; Tessler et al., 2021). We have added text emphasizing this framing (Discussion §1) .
>
> Scaling to more open-ended environments is an important future direction for AI, and we expanded the discussion to note how recent advances in program synthesis and LLM-based code modeling could support this (§6). The fact that humans routinely build rich executable simulations of complex processes (physics engines, game environments, scientific models), suggests that this is both a valuable and a feasible future direction.
>
> **2. Experimental Complexity Limitation**
>
> VGDL games are not high-fidelity environments, but they are well suited for studying human-like inductive structure learning: they require causal inference, multi-step planning, and hypothesis testing, yet are controlled enough for explicit Bayesian inference over executable theories. As our experiments show, these tasks are challenging for deep RL and LLM agents, but they are also challenging to humans: participants often fail to go through all 4 levels of the games within their 15 lives budget. Considering more challenging games (e.g. requiring dozens of hours of game play) would make human experiments considerably more challenging and thus limit the possibilities for model analysis. As discussed in the answer above, scaling our approach to higher-dimensional or continuous domains is an important future direction for AI, but it is orthogonal to our main conceptual contribution: providing a mechanistic account of how structured hypotheses can be jointly updated from language and experience.
>
> **3. Insufficient Comparison to Language Baselines**
>
> Our paper focuses on settings where each participant/model receives a single, novel message written by a previous player. In this regime, existing language-conditioned RL methods are not applicable: these approaches require large amounts of paired (text, episode) experience to learn a mapping from language to policies, and they are not expected to generalize well to entirely new, idiosyncratic messages. In our setting (seeing a message for the first time), such models would behave effectively like pure deep RL, which is why we did not include them as baselines; we now clarify this point more explicitly in the paper (see Baseline models paragraph).
>
> Our LLM agent baseline already contains all the features the reviewer describes. It leverages a large language model (70B) and combines elements of state-of-the-art approaches such as structured planning with ReAct (Yao et al., 2022), chain-of-thought reasoning (Wei et al., 2021), and planning with intermediate sketches  (Nye et al., 2021), cf prompt in Appendix J. In general, LLMs still do not perform well on long-horizon video games (cf https://balrogai.com/), and VGDL further requires inferring novel dynamics from interaction alone, rather than operating over known rule sets as in Balrog-like benchmarks. In this regime, poor performance from pure LLM planning is therefore expected and highlights the need for structured, executable representations.
>
> **4. Technical Validity of the LLM Probability Approximation**
>
> We agree that the language likelihood P(L|T) is necessarily approximate,  because exact speaker likelihoods are unavailable for human speakers. In Bayesian cognitive modeling, approximate generative models (usually defined by hand) are routinely used to capture how linguistic evidence shifts beliefs across hypotheses.
>
> In our framework, the LLM serves as (1) a flexible proposal distribution over rule edits and (2) an approximate speaker model.  The proposal distribution doesn't need to be extremely accurate, as they will be verified and refined by the inference process. The inference algorithm relies only on relative likelihoods across theories, not absolute probability calibration. Empirically, incorporating the language likelihood improves inference, accelerates learning, and better matches human behavior. We have clarified this point in the text (Section 3, likelihood from language).

---

### Author Response · Authors · 2025-11-20
**General response**

We thank the reviewers for their thoughtful and constructive feedback. We appreciate the strong overall positivity of the reviews and the recognition of the contribution, clarity, and empirical breadth of the work. The comments helped us identify several points that benefited from clarification or deeper contextualization. In the revised draft, we have added explanatory text in the Discussion and Methods sections, clarified the definition of structured world models, expanded the rationale for baseline choices, and added new appendix material (e.g., extended DQN results). Below, we address each point raised by the reviewers in detail.

---

### Comment · Area_Chair_WNcg · 2025-11-26
**Query on participant recruitment and compensation.**

Dear authors,

One of the reviewers noted that "A substantial part of this study is about human behavior. I was looking through the SI and did not find a paragraph about participant recruitment and compensation. And this is quite important to report in the paper of the main contribution comes from observations from behavioral studies."

I may have missed your response to this, or perhaps this was not visible to you.

In any case, it would be good if you can reply to my post with your response.

Thanks,
AC

---

> ### Author Response · Authors · 2025-11-26
> **Human experiments**
>
> Thank you for raising this matter to our attention.
>
> Some details about participant recruitments (number of participants, selection criterion, recruitment platform) were already present in the first version of the paper (Section 4, paragraph *Participants*). This said, the original version did omit a number of details, that we now added to Appendix G now renamed *Human data collection: instructions and participant recruitments.* Participants were all adult English-speakers and the median time spent and median hourly pay (49min, $10/h).

---

### Author Response · Authors · 2025-11-30
**Summary of reviews and responses**

Here is a short summary of the reviews and responses to help the final decision.

The reviewers engaged with the submission and were consistently positive about the clarity of the writing, the conceptual contribution, and the empirical breadth of the work. All reviewers initially recommended acceptance (scores 6–8), and the two reviewers who had time to enter the discussion after our responses indicated that their concerns had been addressed and proposed to raise their scores.

Two main points were raised by multiple reviewers: (i) the scalability and environment complexity, and (ii) the baseline choices. We clarified both in the rebuttal and updated the paper accordingly—strengthening the framing of VGDL as a cognitively grounded hypothesis space, expanding the discussion of scalability (Discussion), and adding justification for baseline selection (Section 3). In particular, we now include extended DQN results (in Appendix) and an explanation of why language-conditioned RL methods are not applicable in our single-message regime.

Other concerns were raised and have also been addressed: we added missing recruitment and compensation details for human experiments in Appendix G (Reviewer 46yR), clarified that the multiple-choice rule interface introduces no information leak (Reviewer LmtC), and corrected misunderstandings about the role of the LLM in the architecture: it is only used as a speaker model and proposal distribution with a approximate inference algorithm (Reviewers GnG7 and 46yR). We added discussion of game-variability in the Result section (Reviewer 46yR).

We believe the discussion phase strengthened the paper. The reviewers’ questions helped us clarify our argument, refine the presentation of the computational model, and add a few missing methodological details. We thank the reviewers for their engagement and believe the revised version addresses all major concerns raised by the initial reviews.

---

### Meta-Review · Area_Chair_wUBj · 2026-01-07

**Summary:**

All reviewers like the novel theoretical approach which treats language as probabilistic evidence for Bayesian inference, and the connections to cogsci. They liked the sample efficiency and learning performance, the experimental design and insights into human-AI collabration.

For weakness, reviewers mentioned the limitation to VGDL game and simple environments (6yR, LmtC, kQQx), limited baseline comparisons and how this can scale to more complex environments.

All reviewers are in agreement to accept the paper.

**Reviewer Concerns:**

all reviewers are in agreement to accept.

**Reviewer Scores:**

N/A

---

### Decision · Program_Chairs · 2026-01-26

Accept (Poster)